# Self-renewing human naïve pluripotent stem cells dedifferentiate in 3D culture and form blastoids spontaneously

Mingyue Guo [1,2,3,4,7] ✉, Jinyi Wu[1,2,3,7], Chuanxin Chen [1,2,3,4,7], Xinggu Wang [2,7], An Gong [1,2,3,4], Wei Guan [1,2], Rowan M. Karvas[5], Kexin Wang[2], Mingwei Min [1,2], Yixuan Wang [6], Thorold W. Theunissen [5], Shaorong Gao [6] & José C. R. Silva [2] ✉

Human naïve pluripotent stem cells (hnPSCs) can generate integrated models of blastocysts termed blastoids upon switch to inductive medium. However, the underlying mechanisms remain obscure. Here we report that self-renewing hnPSCs spontaneously and efficiently give rise to blastoids upon three dimensional (3D) suspension culture. The spontaneous blastoids mimic early stage human blastocysts in terms of structure, size, and transcriptome characteristics and are capable of progressing to post-implantation stages. This property is conferred by the glycogen synthase kinase-3 (GSK3) signalling inhibitor IM-12 present in 5iLAF self-renewing medium. IM-12 upregulates oxidative phosphorylation-associated genes that underly the capacity of hnPSCs to generate blastoids spontaneously. Starting from day one of self-organization, hnPSCs at the boundary of all 3D aggregates dedifferentiate into E5 embryo-like intermediates. Intermediates co-express SOX2/OCT4 and GATA6 and by day 3 specify trophoblast fate, which coincides with cavity and blastoid formation. In summary, spontaneous blastoid formation results from 3D culture triggering dedifferentiation of hnPSCs into earlier embryo-like intermediates which are then competent to segregate blastocyst fates.

Deciphering human early embryo cell fate specification is one of the most fascinating research topics in development biology. Nonetheless, limited access to embryos and ethical issues are obstacles to our better understanding of early human development. Single-cell sequencing and transcriptomic analysis proposed different models as to how human pre-implantation embryo lineages specify[1–3]. Recent studies suggested a conserved lineage specification process in the pre-implantation embryo between human and mouse[4–6]. During early embryo development, outer trophectoderm (TE) and inner cell mass (ICM) cells are segregated first, and subsequently ICM cells segregate at the blastocyst stage into naïve pluripotent epiblast (EPI) and hypoblast (HYP)[7–9].

Blastoids are emerging models for early embryo development exploration in vitro. While both embryonic and extraembryonic stem cells are required for embryo-like structure assembly in mouse[10–16] and bovine[17], integrated human blastoids can be generated solely from

[1]Guangzhou Medical University, Guangzhou, Guangdong, China. [2]Guangzhou National Laboratory, Guangzhou International Bio Island, Guangzhou 510005 Guangdong, China. [3]Bioland Laboratory, Guangzhou International Bio Island, Guangzhou 510005 Guangdong, China. [4]The Fifth Affiliated Hospital of Guangzhou Medical University, Guangzhou 510700, China. [5]Department of Developmental Biology and Center of Regenerative Medicine, Washington University School of Medicine, St. Louis, MO 63110, USA. [6]Shanghai Key Laboratory of Maternal and Fetal Medicine, Clinical and Translational Research Center of Shanghai First Maternity and Infant Hospital, School of Life Sciences and Technology, Tongji University, Shanghai 200092, China. [7]These authors contributed equally: Mingyue Guo, Jinyi Wu, Chuanxin Chen, Xinggu Wang. ✉e-mail: guo_mingyue@gzlab.ac.cn; jose_silva@gzlab.ac.cn

hnPSCs using instructive media[18–21]. Human blastoids formed by reprogramming intermediates from adult somatic cells could not accurately mimic blastocyst-stage TE cell fate, whereas those induced from hnPSCs were generally recognized as remarkable reproductions of natural human blastocysts[22,23]. Human naïve embryonic stem cells (hnESCs) can be induced to differentiate into trophoblast fate and into extraembryonic endoderm fate with relative efficiency and this may be linked to the capacity of hnPSCs, upon instruction, to generate blastoids that recapitulate the cellular organization and lineage composition of human blastocysts[24–28]. It should be noticed that ERK, HIPPO, NODAL pathways inhibitors and other TE/HYP inducing factors are used in blastoid inductive culture conditions[18–21]. In this study, however, we found that human blastoids can be spontaneously generated during routine self-renewing culture. All that was required was for hnPSC aggregates to grow in suspension. Having a defined system in which no change to medium composition is employed led us to then investigate what causes blastoid formation and how these are formed starting from cells with naive pluripotent identity. The resultant findings provide insights into the biology of blastoid formation and highlight an inherent plasticity to hnPSCs, which in suspension can dedifferentiate and subsequently segregate blastocyst-like cell fates.

## Results

### Spontaneous emergence of blastoids in self-renewing hnESC cultures

While culturing male TJ-1# human naïve embryonic stem cells[29], hereafter hnESCs, in standard 5iLAF self-renewing medium[30] we frequently noticed that suspended cyst structures appear in cultures on day 2 post-passaging and that by day 3 the size of these increased ($59.6 \pm 10.6 \, \mu m$) (Fig. 1a, Supplementary Fig. 1a, b). The cavity containing cysts consisted of a few cells (2–8) that were SOX2$^{high}$GATA6$^{low}$, SOX2$^{low}$GATA6$^{high}$ or SOX2$^{high}$GATA6$^{high}$ (Supplementary Fig. 1c). TE marker GATA3 expression was not found in these cysts. Cysts collapsed gradually from day 4 but, surprisingly, a few blastocyst-like structures with cavities were consistently observed on day 6 (Supplementary Fig. 1d). Immunofluorescence staining showed that the blastocyst-like structures were usually composed of inner EPI-like cells (SOX2$^+$GATA6$^-$GATA3$^-$), outer TE-like cells (SOX2$^-$GATA6$^+$GATA3$^+$) and a few putative HYP-like cells (SOX2$^-$GATA6$^+$GATA3$^-$) (Supplementary Fig. 1e). As these blastocyst-like structures emerge while in self-renewing culture medium we termed them spontaneous blastoids.

### Rapid and efficient generation of spontaneous blastoids in 3D culture systems

The above phenomena prompted us to speculate if hnESCs might be poised to generate blastoids spontaneously as long as suspended cell aggregations were formed. To test our hypothesis, we cultured hnESCs in two alternative 3D culture systems, suspension plate or AggreWell, with self-renewing 5iLAF medium. Interestingly, hnESCs self-organized spontaneously, rapidly, efficiently and consistently, into blastoids in both settings (Fig. 1b–d). Generally, suspended cell aggregations formed the day after seeding, and cavitation was evident from day 3 (Fig. 1c–e). We optimized the starting cell number in AggreWell (containing 1200 microwells) and self-organized blastoids appeared rapidly and in over 50% of the microwells when ~30 cells/microwell were seeded (Fig. 1f and Supplementary Movies 1, 2).

Immunofluorescence staining showed that blastoids expressed key blastocyst markers, regardless of whether they were organized in a suspension plates (Fig. 1g) or AggreWells (Fig. 1h–j). While the expression of EPI marker SOX2 was detected in the inner part of blastoids, the TE marker GATA3 was detected in the outer monolayer, and there was no overlap between these two populations, suggesting the specification of distinct cell fates. Other EPI or TE markers such as OCT4, NANOG and KRT18, as well as tight junction marker ZO-1 were also found in the expected spatial locations, simulating the

architecture of the human blastocyst (Fig. 1i-j). Spontaneous blastoids contained comparable number of EPI (SOX2$^+$) and TE (GATA3$^+$) cell analogs to that of human embryos and other human blastoids[20] (Fig. 1k). Importantly, GATA3 expression was not detected in self-renewing hnESCs cultures nor in day 1 suspended hnESC aggregates that lead to subsequent blastoid generation (Supplementary Fig. 1f). This suggests that TE-like cells are specified upon 3D culture and during blastoid formation as opposed to pre-existing TE cell differentiation among starting hnESCs.

Next, we asked if 5iLAF hnESCs could also generate blastoids following previously reported blastoid inducing and organizing procedures[18,20]. These multi-step strategies use stage-specific medium and factors to induce TE and HYP specification, and blastoid assembly and start from hnPSCs cultured in PXGL self-renewing medium. Our results showed that 5iLAF cultured hnESCs were unable to make blastoids following these protocols (Supplementary Fig. 2a) suggesting that some optimization was required. However, when self-renewing in PXGL medium and then transferred into AggreWells with 5iLAF medium, hnESCs generated blastoids spontaneously and efficiently (Supplementary Fig. 2b).

During the development of human early embryos, GATA6 is detected in the early blastocyst, then becomes restricted to HYP, and maintains lower expression in TE of mid-blastocysts[31–33]. Similarly, in most of our spontaneous blastoids, GATA6 was co-expressed with SOX2 in the surface layer of inner populations or with GATA3 in the outer enclosing cell layer (Fig. 1h). Only very few GATA6$^+$SOX2$^-$GATA3$^-$ or GATA6$^+$OCT4$^-$GATA3$^-$ cells (Fig. 1h, i) could be observed in spontaneous blastoids. This implies that our spontaneous blastoids are mimicking early stage human embryos, just following cavitation before the complete downregulation of pluripotent markers in the HYP cells.

To functionally characterize spontaneous blastoids, we asked if early embryo lineage representatives could be readily derived from these. When blastoids were individually plated on inactivated mouse embryonic fibroblast cells with corresponding TSM[34], RACL/NACL[28] or 5iLAF[30] medium, human trophoblast stem cells (TSCs, TP63$^+$GATA3$^+$CDX2$^-$), naïve extraembryonic endoderm (nEnd, PDGFRA$^+$GATA6$^+$NANOG$^-$) and blastoid-hnPSCs (OCT4$^+$NANOG$^+$) were derived, respectively (Supplementary Fig. 2c–l). The derived TSC-like cells expressed key genes of trophoblast (Supplementary Fig. 2d, f) and possessed bipotential to differentiate into extravillous trophoblast (EVT)-like cells and multinucleated syncytiotrophoblast (ST)-like cells (Supplementary Fig. 2g, h) using published methods[34]. Blastoid-nPSCs were able to generate secondary blastoids spontaneously and the efficiency was comparable to that of primary hnESCs (Supplementary Fig. 2m, n).

We next asked about the culture requirements for spontaneous blastoid formation. Our results showed that removing individual 5iLAF medium factors dramatically reduced the efficiency of spontaneous blastoid generation (Fig. 1l, m). Importantly, apart from ROCK inhibitor (ROCKi), the absence of individual 5iLAF components impaired blastoid formation efficiency without affecting cell proliferation (Fig. 1m, n). ROCKi Y-27632 is known to promote cell survival, suggesting that cell viability upon hnESC dissociation is not only a crucial parameter in our spontaneous blastoids but also in other blastoid systems[21,35]. These results further demonstrate that robust and complete self-renewing medium can support blastoid formation.

In summary, we found that embryo-like structures, expressing key lineage markers of early-stage human blastocysts, could be readily and spontaneously formed from hnESCs in 3D culture with standard 5iLAF self-renewing medium.

### GSK3 signaling inhibitor switch enhances OXPHOS in hnESCs

Parental TJ-1# hnESCs, the cell line being used in this study, were derived in 5iLAF medium in which the GSK3 signaling inhibitor used

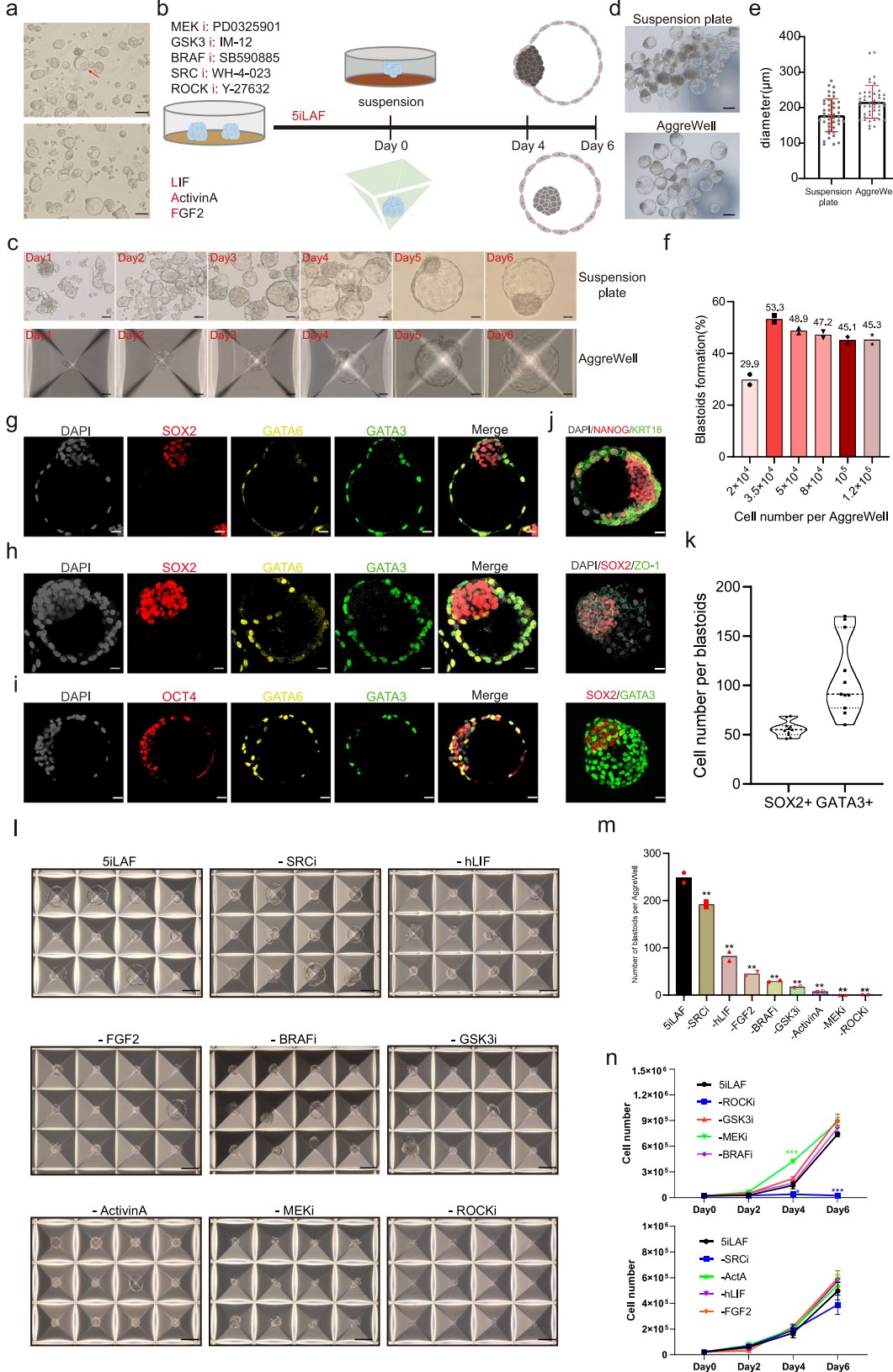

was CHIR99021 (5iLAF-C)[29]. GSK3 inhibitor is one of the 5 inhibitors of signaling pathways used in the 5iLAF medium to maintain human naïve pluripotent stem cells self-renewing and in its original formulation uses IM-12 as GSK3 inhibitor[30]. We replaced CHIR99021 with alternative GSK3 inhibitor IM-12 (5iLAF-I) as this switch observably improved the morphology of our hnESCs (Fig. 2a). On the other hand,

hnESCs self-renewing in 5iLAF-C medium, hereafter C-hnESCs, exhibited better cell proliferation compared to hnESCs in which the GSK3 inhibitor was switched to IM-12, hereafter CI-hnESCs (Fig. 2b). Both CI-hnESCs and C-hnESCs express comparable levels of naïve markers (KLF17, SUSD2, ALPG) to other hnESC lines (Fig. 2c). Interestingly, neither C-hnESCs, derived and maintained with 5iLAF-C medium, nor

**Fig. 1 | Self-renewing hnESCs form blastoids spontaneously. a** Representative phase-contrast images of spontaneous cysts (red arrow) in the hnESCs cultures. Scale bars, 50 µm. (*n* > 10). **b** Schematic of spontaneous blastoids formation from hnESCs in suspension plates (top) and in AggreWell (bottom) with 5iLAF medium (components are listed). Created in Adobe Illustrator. **c** Representative phase-contrast images of blastoids at indicated times points in suspension plates (top) and in AggreWells (bottom). Scale bars, 50 µm. (*n* > 10) **d** Representative phase-contrast images of day6-blastoids formed spontaneously in suspension plate (top) and AggreWells (bottom). Scale bars, 200 µm. **e** Quantification of diameter of day6-blastoids (suspension plate, *n* = 43(number of blastoids measured), 178.5 ± 46.5 µm; AggreWell, *n* = 47, 216.2 ± 46.1 µm; mean ± s.d.). **f** Spontaneous blastoid formation efficiency related to starting cell number per AggreWell. *n* = 2

(technical replicates). **g–j** Representative immunofluorescence staining images of blastoids formed in suspension plate (**g**) or in AggreWell (**h–j**). Scale bars, 20 µm. **k** Quantification of SOX2 and GATA3 cell number per blastoid. *n* = 11 blastoids. **l** and **m**, blastoid generation efficiency in AggreWells upon removal of single factors of 5iLAF medium. **l** indicates representative images and **m** indicates the number of blastoids generated upon removal of indicated factor. *n* = 2 technical replicates; mean; unpaired two-tailed *t*-test; **P < 0.01. Scale bars, 200 µm. **n** Proliferation of hnESCs cultured in 5iLAF minus each indicated factor. Two rounds independent experiments (top and bottom) with parallel control groups cultured in 5iLAF. *n* = 3 technical replicates; mean ± s.d.; unpaired two-tailed *t*-test; *P = 0.0119; ***P = 0.0003 (-MEKi); ***P = 0.00000139 (-ROCKi). Source data are provided as a Source Data file.

CI-hnESCs, could generate blastoids spontaneously when cultured in 3D with 5iLAF-C medium. Conversely, C-hnESCs and CI-hnESCs generated spontaneous blastoids efficiently and consistently in 5iLAF-I medium (Fig. 2d, Supplementary Fig. 3a). These results indicate that the switch of GSK3 inhibitor CHIR99021 to IM-12 capacitates hnESCs with the ability to spontaneously form blastoids.

To investigate potential underlying molecular differences between C-hnESCs and CI-hnESCs, we performed bulk RNA-seq analysis. Principal component analysis (PCA) showed that CI-hnESCs and C-hnESCs, cultured in 5iLAF-I and 5iLAF-C respectively, were distinguishable transcriptionally. CI-hnESCs, but not C-hnESCs, cluster closely with UCLA20 hnESCs, an independent human hnESC cell line that was derived and maintained in 5iLAF using IM-12 as GSK3 inhibitor[36] (Fig. 2e). Compared to C-hnESCs, oxidative phosphorylation (OXPHOS) related mitochondrial genes *MT-s* and *ATP6VOs* were significantly upregulated in CI-hnESCs (Fig. 2f). Genes involved in cell adhesion and extracellular matrix such as *LAMB3* (laminin subunit Beta 3), *PCDH7* (protocadherin), *PCDH19*, *COL3A1* (collagen), *MMP9* (matrix metallopeptidase) and *FBN3* (fibrillin) were also enhanced in CI-hnESCs (Fig. 2f, g). This agrees with the compacted morphology of the colonies (Fig. 2a) and with a recent study revealing that extracellular matrix sustains naïve identity of hniPSCs[37]. Conversely, developmental and stem cell differentiation related genes (*PECAM1, KIT, HAND1, VASH1, CDX2* and *ADM*) were downregulated in CI-hnESCs (Fig. 2f, g, Supplementary Data 1). Taken together, OXPHOS and expression of extracellular matrix and cell adhesion genes were enhanced in CI-hnESCs.

hnPSCs have the flexibility to adjust their metabolic program in response to changing environmental conditions[38]. Both glycolysis and oxidative respiration are enhanced upon acquisition of naïve pluripotency compared to primed pluripotency, which is mainly glycolytic[39,40]. We asked if OXPHOS is required for CI-hnESCs to generate spontaneous blastoids. To this end, we blocked OXPHOS during spontaneous blastoids formation with the inhibitor IACS-010759 (OXPHOSi)[41]. In agreement with the RNA-seq analysis, the treatment with OXPHOSi prevented the generation of blastoids from CI-hnESCs (Fig. 2h, Supplementary Fig. 3b). We also tested if OXPHOS could be underlying blastoid formation using an independent method[42]. Similarly, we observed OXPHOSi significantly impaired blastoid formation in this system (Supplementary Fig. 3c). Importantly OXPHOS inhibition did not visibly alter hnESC proliferation in 2D culture or aggregate size in 3D culture during blastocyst formation (Fig. 2i, Supplementary Fig. 3b). This indicates that OXPHOS, together with 3D culture, underlies the capacity of hnESCs to undergo spontaneous blastoid formation.

### Spontaneous blastoids resemble human blastocysts

To further molecularly profile spontaneous blastoids, we analyzed the transcriptome of day6-blastoids from AggreWell by single-cell RNA sequencing (scRNA-seq). Uniform manifold approximation and projection (UMAP) analysis showed that several distinct transcriptomic states were present in our blastoids, including NANOG+OCT4+GATA6-GATA2-GATA3low EPI-like cells (cluster 2, 3, 4, 5

and 7), NANOG+OCT4+GATA6+GATA2-GATA3+ intermediates (cluster 8), NANOG-OCT4lowGATA6+GATA2+NR2F2low early TE-like cells (cluster 6), and NANOG-OCT4-GATA6lowGATA2+NR2F2+ late TE-like cells (cluster 1 and 9) (Fig. 3a–c, Supplementary 4a). Higher expression level of IFI16 in cluster 7 indicated a relatively later EPI-like cell fate[4]. Moreover, EPI-like cells of spontaneous blastoids were enriched in naïve pluripotency and blastocyst-stage EPI transcripts such as *DPPA5, DNMT3L, KHDC1L, KHDC3L, FGF4, UTF1, KLF4, KLF17, SOX2* and *PRDM14*. TE-like cells were enriched in early TE markers such as *GATA3, KRT8, KRT18, KRT19* and late TE markers *KRT7, NR2F2* and *CGA*. Cluster 8 GATA6+NANOG+GATA3+ intermediates were enriched in transcripts *DPPA3, TET2* and *ALPG*, the human naïve pluripotency specific marker as identified previously[29] (Fig. 3d, Supplementary 3b, Supplementary Data 2). We noticed that fewer TE-like, compared to EPI-like cells, were captured by scRNA-seq (Fig. 3b). This indicates a technical issue with the capture of TE-like cells as observed in other embryo model studies[20,43].

We then compared the transcriptional signatures of our spontaneous blastoids to published scRNA-seq data from human pre- and post-implantation embryos[3,44,45] and from other blastoids[18,20,22,46] (Fig. 3e, f, Supplementary 4c–i). Blastoid cells in cluster 2, 3, 4, 5 and 7 represented EPI cell fate closely, while cluster 6, 1 and 9 resembled early, medium and late TE cell fate respectively. Cluster 8 cells were closer to human early blastocyst stage and to embryo EPI/HYP intermediates (Fig. 3e–g). Importantly, TE-like, EPI-like and intermediate cells of our blastoids mimicked closely counterpart human embryos and other reported blastoids (Supplementary Fig. 4d–i). Cell-fate trajectory analysis also suggested that the TE-like cells in our spontaneous blastoids were specified from cluster 8 GATA6+ intermediate cells (Fig. 3h). Together, these results demonstrate that spontaneous blastoids recapitulate transcriptional features of human early blastocysts and those of other blastoid models.

### BRAF/MEK signaling promotes HYP specification in spontaneous blastoids

In our spontaneous blastoids scRNA-seq analysis we noticed that HYP-like cells are absent (Fig. 3e, f). Given that ERK phosphorylation is essential for HYP fate in mouse and bovine embryos[47,48] and for HYP-like cell differentiation from hnESCs[28], we asked if inhibitors of BRAF and MEK, upstream kinases for ERK phosphorylation, in 5iLAF medium were preventing HYP specification in our spontaneous blastoids. Hence, we removed BRAF and MEK inhibitors at day 3, when cavitation and TE specification are occurring, and analyzed these two days later at day 5 (Fig. 4a). Spontaneous blastoid efficiency was not altered by the removal of the two inhibitors at day 3 but interestingly, SOX17+GATA4+SOX2-GATA3- expressing cells were now detected and present at the expected location adjacent to EPI (Fig. 4b, c). Single-cell transcriptome comparison of these spontaneous blastoids against human embryo reference data also revealed the presence of a cell cluster (cluster 5) mapping to human hypoblast cells (Fig. 4d). The HYP-like cluster showed expression of HYP-associated genes (*PDGFRA, SOX17, FOXA2, GATA4, GATA6*) and downregulation of EPI-associated

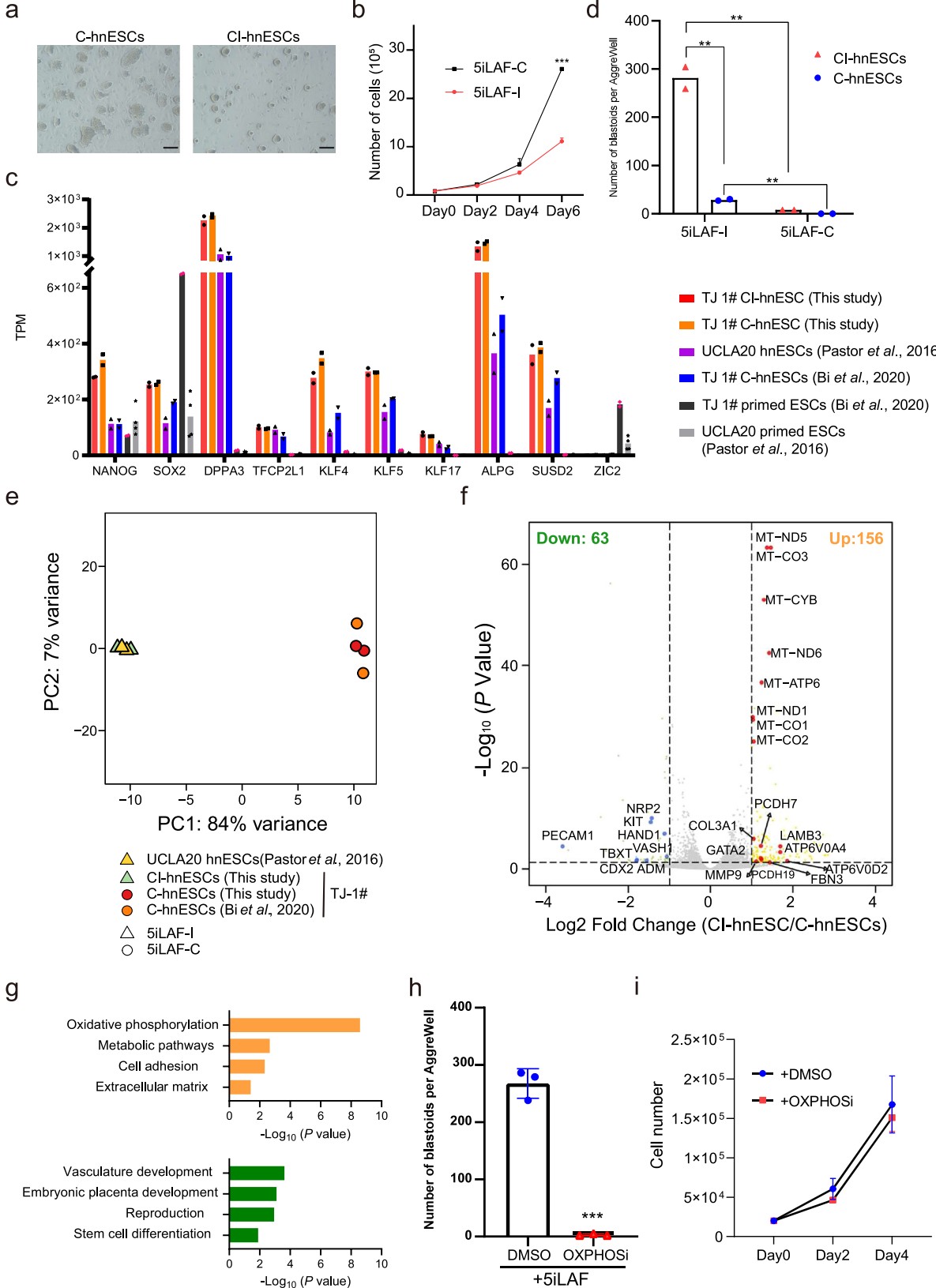

genes (*NANOG*, *SOX2*) (Fig. 4e). Gene ontology (GO) analysis confirmed a plethora of enriched signaling pathways in association with MAPK and BRAF/RAF in the HYP-like cluster (Fig. 4f), suggesting that the absence of HYP specification in 5iLAF spontaneous blastoids was due to the continuous presence of BRAF/MEK inhibition.

Moving day 3 spontaneous blastoids into other RAS-RAF-MEK-ERK inhibition free, non-self-renewing, conditions also led to rapid, within one day, induction of HYP-like cell fate specification (Supplementary Fig. 5a–d). However, in these media blastoid integrity was affected. To attempt improving HYP cell fate specification in our

**Fig. 2 | OXPHOS underlies the capacity of hnESCs to spontaneously generate blastoids. a** Representative phase-contrast images of C-hnESCs (left, cultured in 5iLAF-C, day 6) and CI-hnESCs (right, cultured in 5iLAF-I, day 6). Scale bars, 200 μm. **b** Proliferation of C-hnESCs cultured in 5iLAF-C (black) or 5iLAF-I (red). $n = 3$ replicates; mean + s.d.; unpaired two-tailed $t$-test. ***$p = 0.00000416$. **c** TPM (Transcripts Per Kilobase Million) expression level of representative naïve markers and primed marker ZIC2 among cell lines. $n = 2$ biological replicates ($n = 4$ for UCLA20 primed ESCs). **d** Number of blastoids generated from CI-hnESCs and C-hnESCs with 5iLAF-I or 5iLAF-C medium in AggreWell. $n = 2$ technical replicates; mean; unpaired two-tailed $t$-test; **$P < 0.01$. **e** Principal component analysis (PCA) of bulk RNA-seq datasets from CI-hnESCs, C-hnESCs and published studies. **f** Volcano plots of differential expressed genes identified using two-sided Wilcoxon Rank Sum Test with Bonferroni correction (|Log$_2$ Fold Change | >1 and FDR < 0.05, upregulated genes in yellow and red, downregulated genes in green and blue) in CI-hnESCs comparing to C-hnESCs. **g** Bar plot showing the −log$_{10}$ ($P$ value) of the gene ontology terms enriched in upregulated genes (top) or downregulated genes (bottom). The input genes were selected as was described in **f** and the $P$ values shown were computed with modified one-sided Fisher's Exact test using *David* functional annotation tool. **h** Quantification of blastoids number in 5iLAF-I with dimethyl sulfoxide (DMSO) or OXPHOS inhibitor (IACS-010759, 5 μM). $n = 3$ technical replicates; mean ± s.d.; unpaired two-tailed $t$-test; ***$p = 0.0000607$. **i** CI-hnESC proliferation in 5iLAF-I with DMSO or OXPHOS inhibitor. $n = 3$ replicates; mean ± s.d.; Source data are provided as a Source Data file.

spontaneous blastoid model, we have also removed BRAF/MEK inhibition from day 2. This resulted in a significant increase of HYP cell number, to around 8 HYP cells per blastoid (Supplementary Fig. 5e–i). However, because cavitation in spontaneous blastoids depends on complete self-renewing medium, and this occurs mainly at day 3, removing BRAF/MEK signaling inhibitors at day 2 impacted negatively blastoid generation efficiency (Supplementary Fig. 5h).

Together, these results demonstrate that spontaneous blastoids are competent to specify HYP cell fate.

## Spontaneous blastoids have post-implantation developmental potential

To address if spontaneous blastoids have the potential to progress to post-implantation development, we differentiated day 5 5iLAF spontaneous blastoids for 9 days (14 days in vitro culture in total) using the protocol developed by Karvas et al.[49] with minor modifications. In brief, we simulated natural implantation by culturing our spontaneous blastoids on Cultrex-coated 8-well slides (Fig. 5a). Attachment of the blastoids was rapid as evidenced by the outgrowth of trophoblast-like cells within 24 h (Fig. 5b). These morphological changes indicated blastoids functionally nidate and the EPI lineages differentiate. To delineate the progression of embryonic and extraembryonic lineages of our blastoids, we collected and profiled the differentiated spontaneous blastoids 9 days after attachment at bE14 (Fig. 5c–i and Supplementary Fig. 6). With cross-reference to the pre- and post-implantation embryo data (E3-E19)[3,44,45,50], the bE14 differentiated spontaneous blastoids showed clear alignment to reference post-implantation human embryo lineages and were best matched to the E14-19 dataset (Fig. 5c). Transcriptomic analysis revealed that blastoid mesoderm (bMES), definitive endoderm (bDE) and hemogenic endothelium (bHE) already emerged at the time point which overlapped to E16-19 reference dataset, indicating accelerated progression from primitive streak (bPS) (Fig. 5c and Supplementary Fig. 6a). The faster developmental pace was also observed in other embryo-like model systems[50,51]. Immunostaining confirmed the presence of confined streak-like structures comprising T-expressing cells (Fig. 5e).

HYP derivatives (visceral endoderm/yolk sac endoderm (bVE/YSE) and anterior visceral endoderm (bAVE)) were also identified in our transcriptomic analysis, even though our spontaneous blastoids show impaired HYP commitment until the removal of BRAF/MEK signaling inhibition (Fig. 3). Because bVE/YSE, bAVE and definitive endoderm (bDE) shared a plethora of endodermal markers (GATA4, GATA6, SOX17, LEFTY1) (Fig. 5d and Supplementary Fig. 6b), we questioned the annotation of extraembryonic endoderm lineages. To this end, we downloaded the DEGs between DE and extraembryonic endoderm identities defined in Mackinlay et al.[52] and scored the annotated lineages from our differentiated blastoids against their embryo counterparts. Gene set variation analysis (GSVA) demonstrated that our bDE like cells bore a higher similarity to the embryo DE identity while bVE/YSE and bAVE were more similar to the embryo extraembryonic identities (Fig. 5f). In addition, we found evidence of secreted CER1 proteins in the extracellular regions of GATA4+ cell clusters (Fig. 5g),

suggesting that these cells may potentially be recapitulating the role of AVE cells as a signaling center to counteract BMP signaling[53]. In view of extraembryonic tissue development, we found both trophoblast descendants, cytotrophoblasts (bCTBs) and syncytiotrophoblasts (bSTBs), and amnion-like cells (bAM) present in our UMAP analysis (Fig. 5c and Supplementary Fig. 6c). bAM cells showed highest resemblance to embryo amnion cells (Fig. 5d and Supplementary Fig. 6d). Of note, the reduced number of trophoblast-like cells in sequencing analysis compared to imaging was probably due to a technical issue because these lineages were deeply intruded into the matrix, as a result these cells could be lost while we harvested the samples for sequencing. Yet, bCTBs and bSTBs in our samples showed elevated expression of classic trophoblast markers GATA2, GATA3, NR2F2, TFAP2C (Fig. 5d), and fluorescent staining results showed outgrowth of GATA3+ trophoblast like cells (Fig. 5h, i). Presence of putative bSTBs was supported not only by sequencing results but also by CGB staining (Fig. 5h). Potential EVT precursors were also found as evidenced by the presence of EVT marker HLA-G in GATA3 expressing cells (Fig. 5i). Of note, we did not identify EVTs in the sequencing analysis. However, these are rare at the equivalent embryonic developmental time point[54].

Together, these results suggest that spontaneous blastoids are also a useful post-implantation embryo-like model and confirm that these are competent to segregate extraembryonic endodermal lineages.

## Independent hnPSC lines form spontaneous blastoids

Next we tested spontaneous blastoid competence in two additional human pluripotent cell lines. To convert female STiPS O-XX1 human primed iPSCs[55] into the naïve state, we used 5iLAF-C medium as cell proliferation and primed to naïve conversion is optimized in this medium[29]. A few colonies emerged during the first 12 days, and more dome-shaped C-hniPSCs were observed after passaging. Upon 5iLAF-C to 5iLAF-I medium switch, CI-hniPSC colonies compacted and detached from the feeder layer (Fig. 6a, b). Strikingly, when picked into U-bottom plate, and continually cultured in 3D with 5iLAF-I medium, floating colonies cavitated within 3-4 days generating blastocyst-like structures (Fig. 6b, c). These blastoids, however, had a larger number of cells in the inner/EPI-like compartment (Fig. 6c, d). GATA3 positive cells were present and located in the outer TE-like compartment. To quantify spontaneous blastoid generation efficiency, C-hniPSC and CI-hniPSC single colonies were picked and transferred into individual round bottom wells to be further cultured into either 5iLAF-C or 5iLAF-I. C-hniPSC single colonies were unable to develop blastoids in 5iLAF-C medium but acquired this capacity when placed in 5iLAF-I medium (Fig. 6e, Supplementary Fig. 7a). Furthermore, single cell dissociated CI-hniPSCs were able to generate blastoids in AggreWell albeit cell survival following single-cell dissociation affected efficiency (Fig. 6f, Supplementary Fig. 7b). We also evaluated spontaneous blastoid formation in a third independent and widely used hPSC line, female H9 hESCs. Primed H9 hESCs were first converted into naïve H9 hESCs (H9 hnESCs) as described above.

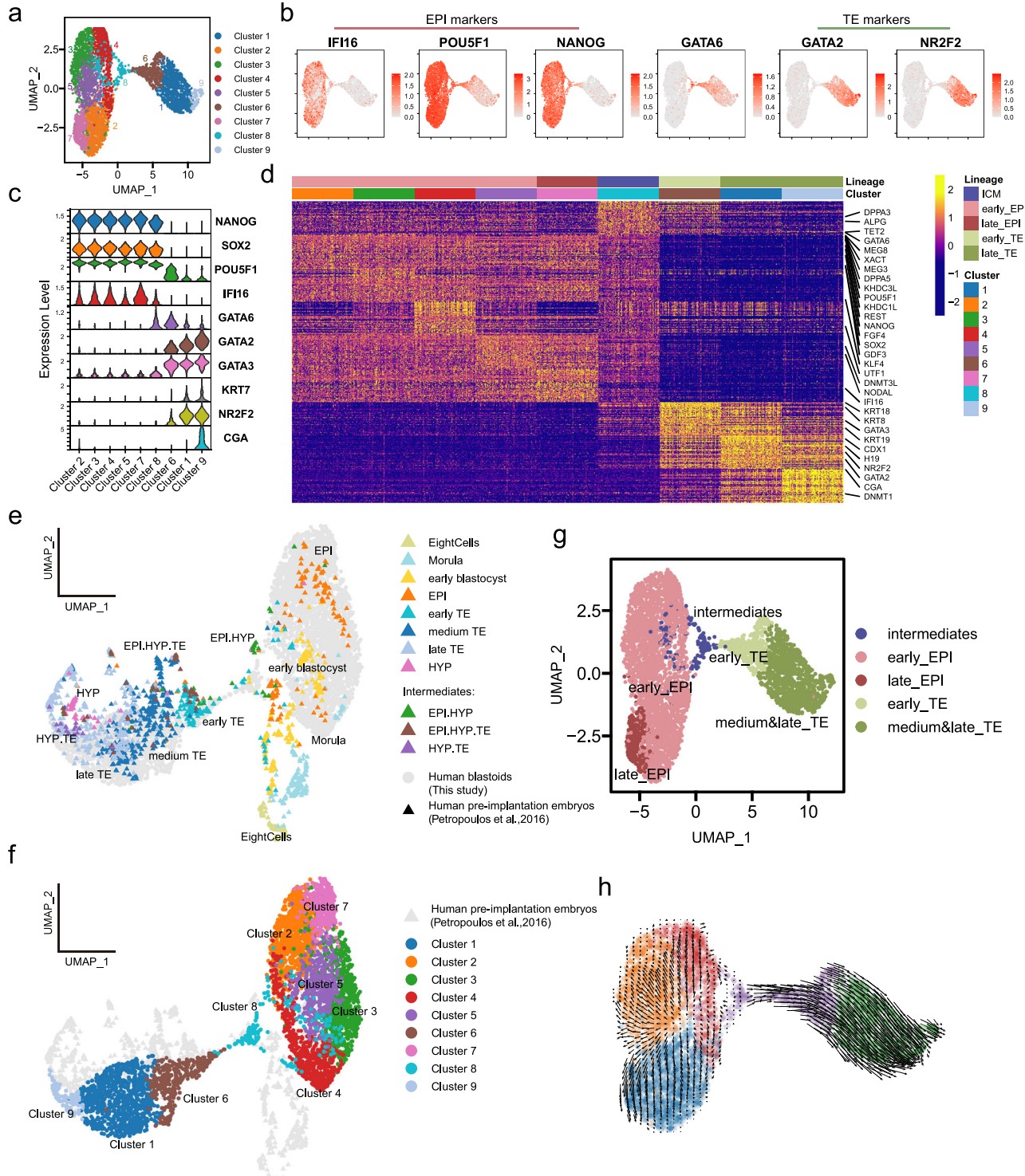

**Fig. 3 | Spontaneous blastoids resemble early human blastocysts. a** Uniform manifold approximation and projection (UMAP) embedding of single-cell transcriptomes from day6-blastoids generated in AggreWell. **b** Expression of human EPI markers IFI16, POU5F1 and NANOG and TE markers GATA2 and NR2F2. **c** Expression level of blastocyst lineage markers in each cluster. **d** Heat map showing the expression levels of the top 50 genes that are enriched in each cluster. Expression levels: z-scores. **e**, **f** UMAP of single-cell transcriptomes of cells from day6-blastoids (**f**) integrated with published datasets from human early embryos (**e**). **g** Rename the clusters (defined in **a** on the basis of similarities to human blastocyst lineages). **h** RNA velocity analysis indicating the cell trajectories shown in arrows.

Consistent with the results for the two other lines, H9 hnESCs also showed competence to form spontaneous blastoids (Supplementary Fig. 7c).

We have also generated hniPSCs, converted from primed pluripotent iPSCs, using exclusively 5iLAF-I medium and without manual colony picking (Fig. 6g). In a side-by-side comparison the hniPSC line showed similar spontaneous blastoid formation efficiency compared to hnESCs (Fig. 6h, k). Importantly, when under the spontaneous blastoid condition for 3 days and then in 3iLAF for 2 days, HYP cell fate specification was also observed (Fig. 6j, k). This further confirms that spontaneously generated blastoids specify HYP fate upon removal of BRAF/MEK signaling inhibition.

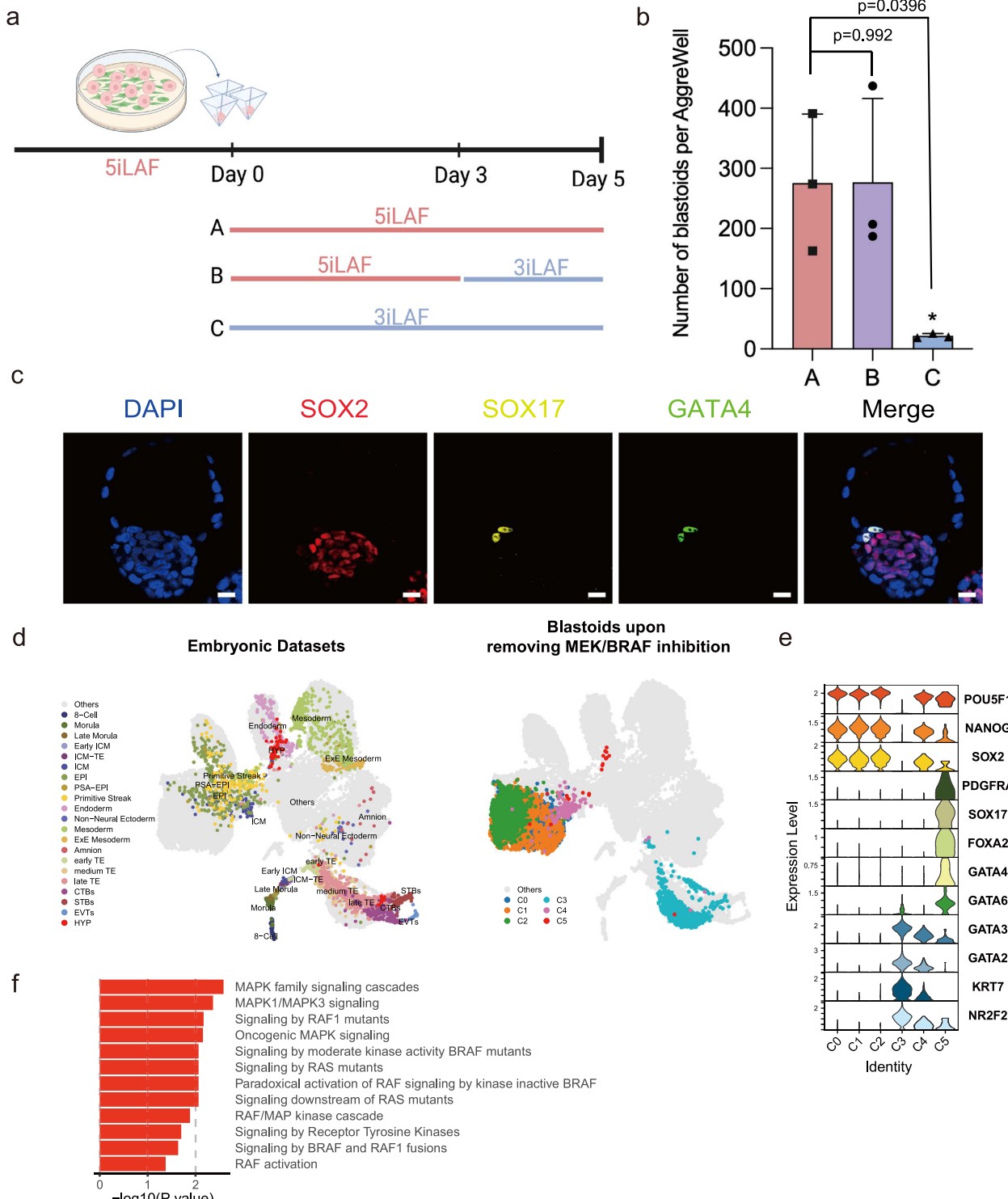

**Fig. 4 | Spontaneous blastoids specify hypoblast fate upon removal of BRAF/ MEK signaling inhibition. a** Schematic of HYP-like cell fate induction by removing BRAF/MEK inhibitors. Created with BioRender.com. Strategies were indicated as follows: A, Spontaneous blastoids were formed by culturing in 5iLAF for 5 days; B, Spontaneous blastoids were formed by culturing in 5iLAF for 3 days first and then switched to 3iLAF (5iLAF-BRAFi-MEKi) for 2 days; C, Spontaneous blastoids were formed by culturing in 3iLAF for 5 days. **b** Spontaneous blastoid generation efficiency for the indicated conditions in **a**. $n = 3$ biological replicates; mean ± s.d.; Ordinary one-way ANOVA with Tukey's multiple comparisons test; * means $p < 0.05$. **c** Representative immunofluorescence staining images of spontaneous blastoids containing HYP-like cells. Scale bars, 20 μm. **d** Comparison of embryonic datasets (left) with the spontaneous blastoids (right) after removing BRAF/MEK inhibitors from day 3 evidenced the HYP fate specification. PSA-EPI, primitive streak anlage stage epiblast. ExE extraembryonic, CTB cytotrophoblasts, STB syncytiotrophoblasts, EVT extravillous cytotrophoblasts. **e** Expression levels of blastocyst lineage markers for each cluster. **f** Enriched gene ontology (GO) terms in the differentially expressed genes of C5 compared against C0-3 identified using two-sided Wilcoxon Rank Sum Test with Bonferroni correction (Log2 Fold Change >0 and FDR < 0.05). The *P* values shown were computed with modified one-sided Fisher's Exact test using *David* functional annotation tool. Source data are provided as a Source Data file.

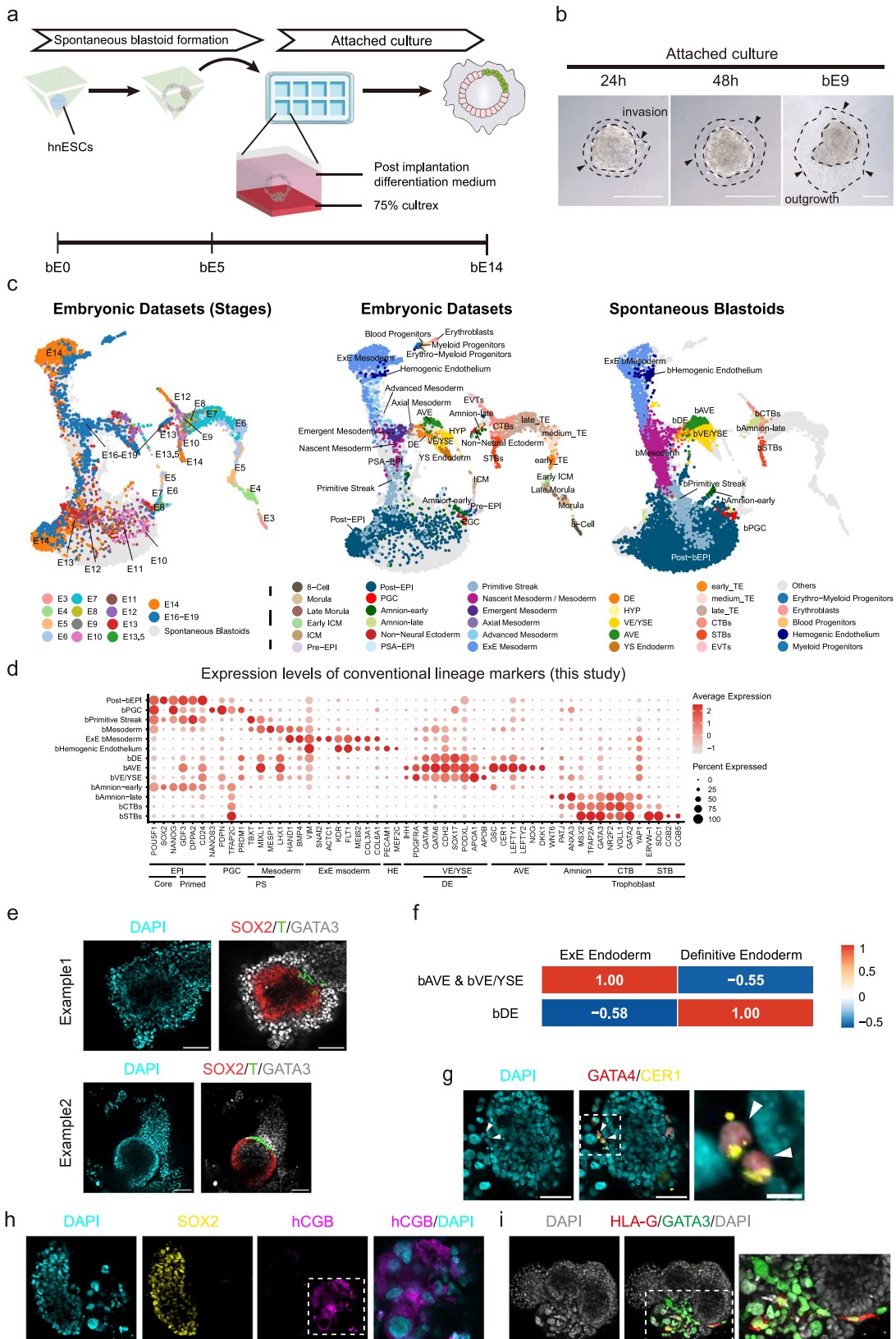

**Nature Communications** | (2024)15:668

Together, these results demonstrate that independent hnPSC lines are also competent to generate blastoids spontaneously.

**Cells at the boundary of 3D hnESC aggregates dedifferentiate**

Next we asked how hnESC aggregates spontaneously generate blastoids. To address this we collected 5iLAF 3D cultured aggregates/

blastoids for immunofluorescence time-course analysis. Our results showed that, starting from day one, GATA6⁺ cells appear at the boundary of hnESC aggregates (Fig. 7a). These also express naïve pluripotency markers and are negative for the TE marker GATA3. At day 2, however, GATA6⁺ cells also start to become GATA3⁺ and SOX2⁻, suggesting TE specification is occurring. By day 3 we observed

**Fig. 5 | Spontaneous blastoids have post-implantation developmental potential. a** Schematic of attached culture of spontaneous blastoids for post-implantation development. Created in Adobe Illustrator. **b** Representative images showing rapid invasion and outgrowth of blastoid trophoblasts. (*n* > 10). **c** Comparison of embryonic datasets (left and middle) with the differentiated spontaneous blastoids on bE14 (right). **d** Gene expression patterns of conventional lineage markers in clusters shown in Fig. 5a. Dot size represents the percentage of cells that express the indicated gene and color intensity represents the expression levels. **e** Representative immunofluorescent images showing two examples of T⁺ primitive streak like structures in bE14 blastoids. **f** Gene set variance analysis (GSVA)

comparing similarities of clusters bAVE, bVE/YSE and bDE to natural embryo ExE endoderm and definitive endoderm. Higher values represent closer similarity. **g** Representative immunofluorescent images of bAVE-like cells showing GATA4 expression in the nucleus and secreted CER1 proteins in the extracellular region. (*n* = 2 from 2 individual experiments). **h** Representative immunofluorescent images of bSTBs showing hCGB expression. (*n* = 2 from 2 individual experiments) **i**, Representative immunofluorescent images of EVT precurosrs showing co-expression of HLA-G and GATA3 (*n* = 2 from 2 individual experiments). All scale bars, 100 μm.

cavitation in the majority of aggregates and this seemingly occurred in those with better GATA3⁺ versus EPI cell ratio (Fig. 7a).

To further characterize how spontaneous blastoids are generated and to understand the undergoing cell fate transitions, we performed time-course scRNA-seq analysis. Using human embryo and human pluripotent stem cell data as reference, we confirmed that our starting hnESCs were similar to E6 human embryo naïve EPI and to reference human naïve ESCs (Fig. 7b, c). Strikingly, day2 3D-cultured hnESC clusters exhibit the presence of cells that have molecularly reprogrammed to an E5 embryo-like intermediate (Fig. 7b). These intermediates are positive for GATA6 expression, which means that they correspond to the GATA6⁺ cells that have emerged at the boundary of hnESC aggregates (Fig. 7a–e). At day 3 and coinciding with cavity formation, E5 embryo-like intermediates are seen undergoing TE cell fate specification as defined by the acquisition of GATA3 and GATA2 expression and the loss of OCT4, NANOG and SOX2 expression (7b-d). Together, these results demonstrate that 3D culture triggers self-renewing hnESCs at the boundary of aggregates to undergo reprogramming to an earlier stage of embryonic development. These are then competent to undergo TE cell fate specification and to induce cavity formation.

## Discussion

We demonstrated that robustly self-renewing hnPSCs give rise to blastocyst-like structures rapidly and efficiently in a spontaneous manner upon 3D culture mode. Importantly, spontaneous blastoids recapitulated human early blastocysts following cavitation and were capable of progressing to post-implantation stages upon attachment to an appropriate matrix that simulates natural implantation. Previous protocols to induce human blastoids differ from ours as they require change of the medium composition into non-self-renewing culture conditions, meaning that medium-mediated instructions were used[18–21,55].

In 5iLAF medium containing the GSK3 inhibitor IM-12, hnPSCs colonies exhibit a compact morphology and reduced expression of differentiation markers compared to the same medium containing instead the GSK3 inhibitor CHIR99021. This is consistent with the capacity of hnPSCs to give rise to blastocyst-like structures being linked to robust self-renewal rather than to pre-existing trophoblast differentiation within hnESC cultures. In 5iLAF medium containing IM-12, hnPSCs express higher levels of OXPHOS-associated genes and we found this to be essential for hnPSCs' ability to spontaneously generate blastoids. Interestingly, gene ontology analysis of single-cell RNA transcriptomes of human in vitro fertilized pre-implantation embryos showed that at the morula and blastocyst stages mitochondria/oxidative phosphorylated genes are among the most significantly upregulated group of genes[1,56]. Oxygen consumption in the early human embryo at the morula stage was also found to correlate with faster progression to the blastocyst stage[57,58] and blastocysts consume high levels of oxygen mediated by high expression of OXPHOS genes[59].

In our study we found that, when in suspension, the surrounding cells of hnESC aggregates dedifferentiate to an earlier stage of embryonic development; that is, they reprogram from an initial E6 embryo-like stage into an E5 embryo-like stage with intermediate

molecular signatures. The intermediates in our spontaneous blastoids were defined by the acquisition of GATA6 expression while remaining OCT4-positive. At day 3 some intermediate cells are seen to have specified TE fate. Interestingly, double positive GATA6 and OCT4 cells were recently identified in the mouse system as having the capacity to give rise to all 3 blastocyst cell fates[60,61]. The further molecular and functional characterization of these GATA6-positive cells and its potential parallels to mouse equivalent cells will therefore be of great interest for future studies.

Trophoblast specification, as marked by the acquisition of GATA3 expression, is well-defined at day 3 in 3D culture and this coincides with the occurrence of cavitation. It will now be interesting to determine if TE cell fate specification is the cause leading to cavitation or if this is just a correlation. We also observed that specification of TE fate occurs in all hnESC aggregates in the surrounding cells; however, cavitation efficiency was on average 50%. Whether this is due to the GATA3⁺ versus EPI cell ratio was not fully determined in this study, but if this is the case it implies that with further optimization it will be possible to achieve an even higher blastoid formation efficiency in this system.

Blastoid systems as a whole show an underrepresentation of the HYP lineage[62] and in our model this was inhibited by the culture conditions supporting spontaneous blastoid formation. 5iLAF medium contains BRAF/MEK inhibitors, which are antagonistic to HYP specification[63]. Refining our model system, in order to combine spontaneous blastoid formation with efficient HYP specification, may improve this as a post-implantation embryo-like model.

The spontaneous formation of embryo-like structures during routine self-renewing culture of human naïve pluripotent stem cells also raises awareness about the culture medium to maintain these. If preventing the occurrence of blastocyst-like structures is needed as far as human embryo models are concerned, the use of GSK3 inhibitor CHIR99021, in detriment of IM-12, is recommended.

In conclusion, our study reports on the spontaneous generation of blastoids and on this being the result of an uncovered direct functional property of hnPSCs. The latter is characterized by hnESCs undergoing dedifferentiation upon switch from 2D to 3D culture, which renders the resultant cells competent to specify blastocyst fates and to induce the formation of a blastocyst structure. This not only changes our conceptual understanding of hnPSCs but also sets up a new platform with which to explore human embryo development in vitro.

## Methods
### Ethics statement
All human ESC, iPSC and blastoid experiments were performed at the Center for Cell Lineage and Atlas (CCLA) of Bioland Laboratory and Guangzhou National Laboratory and followed the 2016 and 2021 Guidelines released by the International Society for Stem Cell Research (ISSCR). Human ESC, iPSC and blastoid work was reviewed and approved by the Ethics Committee of Guangzhou National Laboratory. This work did not exceed a developmental stage normally associated with 14 consecutive days in culture of blastoids as far as human embryo models are concerned.

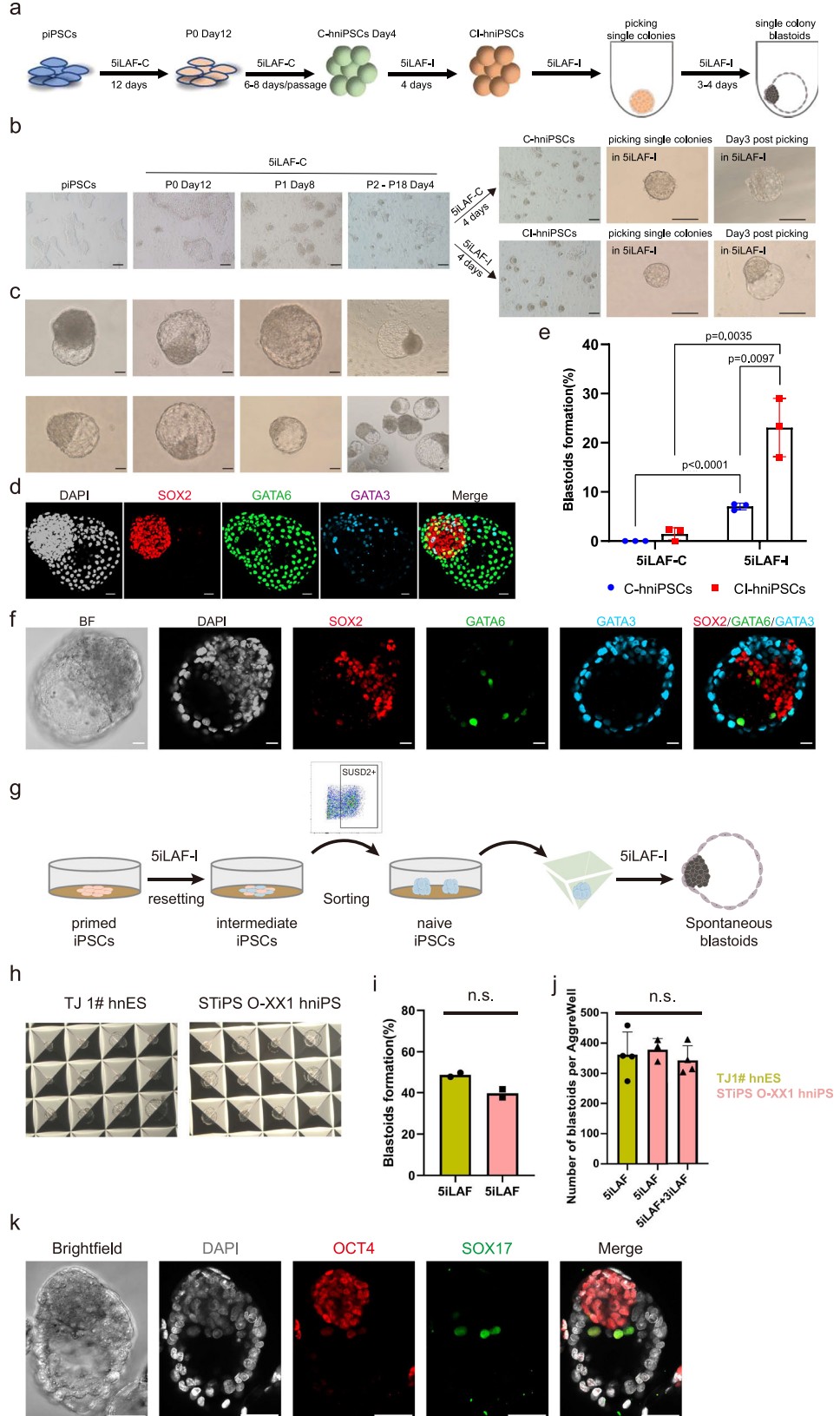

## Culture of human naive PSCs

TJ-1# hnESCs (obtained from Dr. Shaorong Gao and Dr. Yixuan Wang), which are male, were converted from the corresponding primed ESCs as described previously[29]. TJ-1# hnESCs (C-hnESCs) and C-hniPSCs were cultured in 5iLAF-C medium containing 1:1 of DMEM/F-12 (Gibco, C11330500CP) and Neurobasal (Gibco, 21103049), 1% N2 supplement (Gibco,17502048), 2% B27 supplement (Gibco, 17504044), 0.5% KnockOut SR (Gibco, 10828010), 1 mM GlutaMAX (Gibco, 35050061), 1% MEM Non-Essential Amino Acids (Gibco, 11140050), 1% penicillin-streptomycin (Gibco, 10378016), 0.1 mM 2-mercaptoethanol (Gibco, 21985023), 50 μg/ml bovine serum albumin (BSA, Sigma, B2064), and the following small molecules and cytokines: 1 μM PD0325901 (Selleck,

**Fig. 6 | Spontaneous blastoid generation in independent hnPSC lines.**
**a** Schematics of generation of spontaneous individual colony blastoids from iPSCs with GSK3 inhibitor switch strategy. Created in Adobe Illustrator. **b** Representative phase-contrast images of cell, colonies or blastoids during the conversion from piPSCs to hniPSCs, the switch of GSK3 inhibitors and the generation of individual colony blastoids. Scale bars, 200 μm. **c** Representative phase-contrast images of spontaneous blastoids generated from CI-hniPSCs in **b**. Scale bars, 50 μm. **d** Representative immunofluorescence staining images of blastoids in **c**. Scale bars, 50 μm. **e** Quantification of formation efficiency of individual colony blastoids generated from C-hniPSCs and CI-hniPSCs with 5iLAF-I or 5iLAF-C medium in AggreWell. *n* = 3 technical replicates; mean ± s.d.; unpaired two-tailed *t*-test; **P < 0.01, ***P < 0.001. **f** Representative immunofluorescence staining images of blastoids generated from dissociated hniPSCs in AggreWell (Supplementary

Fig. 5e). BF, bright field. Scale bars, 20 μm. **g** Schematics of spontaneous blastoid formation from hniPSCs converted from primed iPSCs using exclusively 5iLAF-I medium and without manual colony picking. Created in Adobe Illustrator. **h** Representative phase-contrast images of blastoids in AggreWell parallelly generated via method from Fig. 1a and 6g. **i** Spontaneous blastoid formation efficiency related to cell line per AggreWell. *n* = 2 (technical replicates); ordinary one-way ANOVA. **j** Quantitative scoring of number of spontaneous blastoids per AggreWell of independent cell lines that were also generated using different methods, *n* = 4 (technical replicates); mean ± s.d; ordinary one-way ANOVA with Tukey's multiple comparisons test. **k** Representative immunofluorescent images of blastoid from hniPSCs in which BRAF/MEK inhibition was removed for 2 days after 3 days in 5iLAF medium. Please note HYP-like SOX17+ cells lining between EPI-like cells and cavity. Scale bars, 100 μm. Source data are provided as a Source Data file.

S1036), 1 μM CHIR99021 (Selleck, S2924), 0.5 μM SB590885 (Selleck, S2220), 1 μM WH-4-023 (Selleck, S7565), 10 μM Y-27632 (Selleck, S1049), 20 ng/ml recombinant human LIF (Qkine, Qk036), 20 ng/ml Activin A (Qkine, Qk001), 8 ng/ml FGF2 (Wuhan Healthgen Biotechnology Corp., HYC005M01). TJ-1# CI-hnESCs, H9 CI-hnESCs and CI-hniPSCs were cultured in standard 5iLAF medium in which 1 μM IM-12 (Selleck, S7566) was used to replace CHIR99021. All nPSCs were cultured on Mitomycin C (Sigma, M0503) inactivated MEF feeders with daily refreshed medium under 20% $O_2$ and 5% $CO_2$ at 37 °C and passaged with Accutase (Invitrogen,00455556). C-hnESCs were passaged every 4-5 days. CI-hnESCs were passaged every 6 days. CI-hniPSCs and C-hniPSCs were passaged every 6-8 days.

## Culture of human primed PSCs
STiPS O-XX1 primed iPSCs[55] (obtained from Dr. Miguel A. Esteban) and H9 primed ESCs (conserved under Dr. Jose C.R. Silva's lab), which are both female, were cultured in Matrigel (Corning, 354277) or Geltrex (Gibco, A1413302) coated plate in mTeSR1 medium (Stemcell, 85850) under 20% $O_2$ and 5% $CO_2$ at 37 °C and passaged with Accutase every 3 days.

## Generation of human blastoids in AggreWell
CI-hnESCs were treated with Accutase at 37 °C for 5 min and neutralized with DMEM/F-12, followed by gentle mechanical dissociation with a pipette. After centrifugation, the cell pellet was resuspended in 5iLAF-I medium and then transferred onto gelatin-coated plate and incubated at 37 °C for one hour to remove feeder cells. After feeder cell adhesion, the supernatant was collected and the cell number was determined using Countstar Mira FL with AOPI fluorescence staining to assess cell viability. $3.5 \times 10^4$ cells were resuspended in 1 ml of 5iLAF-I medium and seeded into one well of an AggreWell 400 24-well plate (Stemcell,34411) pretreated with anti-adherence rinsing solution (Stem Cell Technologies, 07010) following the instructions and cultured under 20% $O_2$ and 5% $CO_2$ at 37 °C. Aggregates formed the other day, and the medium was replaced every day. Usually, spontaneous cavitation took place gradually from day3. To block OXPHOS when generated blastoids, 5 μM OXPHOS inhibitor IACS-010759(Selleck, 8731) was applied once cells were seeded in AggreWell. To induce HYP-like cells, SB590885 and PD0325901 were removed from day 3 or change medium to HDM[19] or IVC[64].

## Generation of human blastoids in suspension plate
$4 \times 10^4$ CI-hnESCs were seeded on 24-well suspension plate (Greiner, 662102), which had been coated with Mitomycin C inactivated MEF feeders (-$1.25 \times 10^5$ cells) for 24 h (plates were not coated with gelatin). CI-hnESCs colonies were always suspended after seeding even there was a webbed feeder layer. Cells were supplied with additional 500 μl of fresh 5iLAF-I medium daily, with the medium was already in the plates. The spontaneous blastocyst-like structures with cavities emerged from day4 to day6. In the suspension plate condition feeder

cells were found to enhance blastoid generation. Cells and blastoids were cultured under 20% $O_2$ and 5% $CO_2$ at 37 °C.

## Generation of blastoids from individual hniPSC colonies
The conversion of primed iPSCs to naïve iPSCs was performed as previous described[29,65]. Briefly, $2 \times 10^4$ STiPS O-XX1 primed iPSCs were seeded on feeders in mTeSR1 medium supplied with 10 μM Y-27632 and cultured under 20% $O_2$ and 5% $CO_2$ at 37 °C. 24 h later, medium was changed to 5iLAF-C and this constituted day0 of the conversion. With daily refreshed medium, a few dome-shaped colonies emerged from day10 to day12 but most of colonies were proliferated in a flat morphology.

Up to day12 when it reaching ~90% confluence, cells were passaged to a fresh plate coated with feeders at a ratio of 1:2 using Accutase. More dome-shaped colonies occurred after the first passage and the morphology got better from the second passage. On day3-4 of every passage (till P18 as we have tested), the medium was changed to standard 5iLAF-I to induce CI-hniPSCs, and constituted day0 of the switching. On day4 suspended CI-hniPSCs detached from the feeders and individual colonies were picked into 96-well U-bottom cell culture plate (Corning, 3799) with 150 μl of 5iLAF-I medium. Fresh medium was topped up to each well the other day. 3-4 days after picking, spontaneous blastocyst-like structures with cavities were observed.

## Generation of blastoids from hniPSC line in 5iLAF-I alone
The conversion of primed iPSCs to naïve iPSCs was performed as previous described with minor modifications[29,65]. Briefly, $1 \times 10^4$ STiPS O-XX1 primed iPSCs were seeded on feeders in mTeSR1 medium supplied with 10 μM Y-27632 and cultured under 20% $O_2$ and 5% $CO_2$ at 37 °C. 24 h later, medium was changed to 5iLAF-I and this constituted day0 of the conversion. With daily refreshed medium, cells were sorted against naïve surface marker SUSD2 (Biolegend, 327406) on day10. The sorted SUSD2+ cells were re-plated at $1 \times 10^4/cm^2$ on feeders and routinely cultured in 5iLAF-I. After 3-5 passages, hnPSCs were ready for spontaneous blastoid formation in AggreWell and further analysis.

## Quantification of blastoid formation efficiency
For AggreWell, the day5 or day6 blastocyst-like structures that contain an ICM-like compartment and a transparent well-defined cavity were counted as blastoids. The blastoid formation efficiency was calculated as the number of blastoids per number of total aggregates in one well of the AggreWell. In some cases, we only counted the number of blastoids because the total number of the microwell in each AggreWell is a determined and fixed value.

For suspension plates, day6 blastocyst-like structures that contain an ICM-like compartment and a transparent well-defined cavity were counted as blastoids. The numbers of blastoids from CI-hnESCs varied from 15 to 50 depending on the webbed feeders and suspended state of CI-hnESCs colonies. Due to the difficulties in segregating and harvesting blastoids formed spontaneously in suspension plates from the

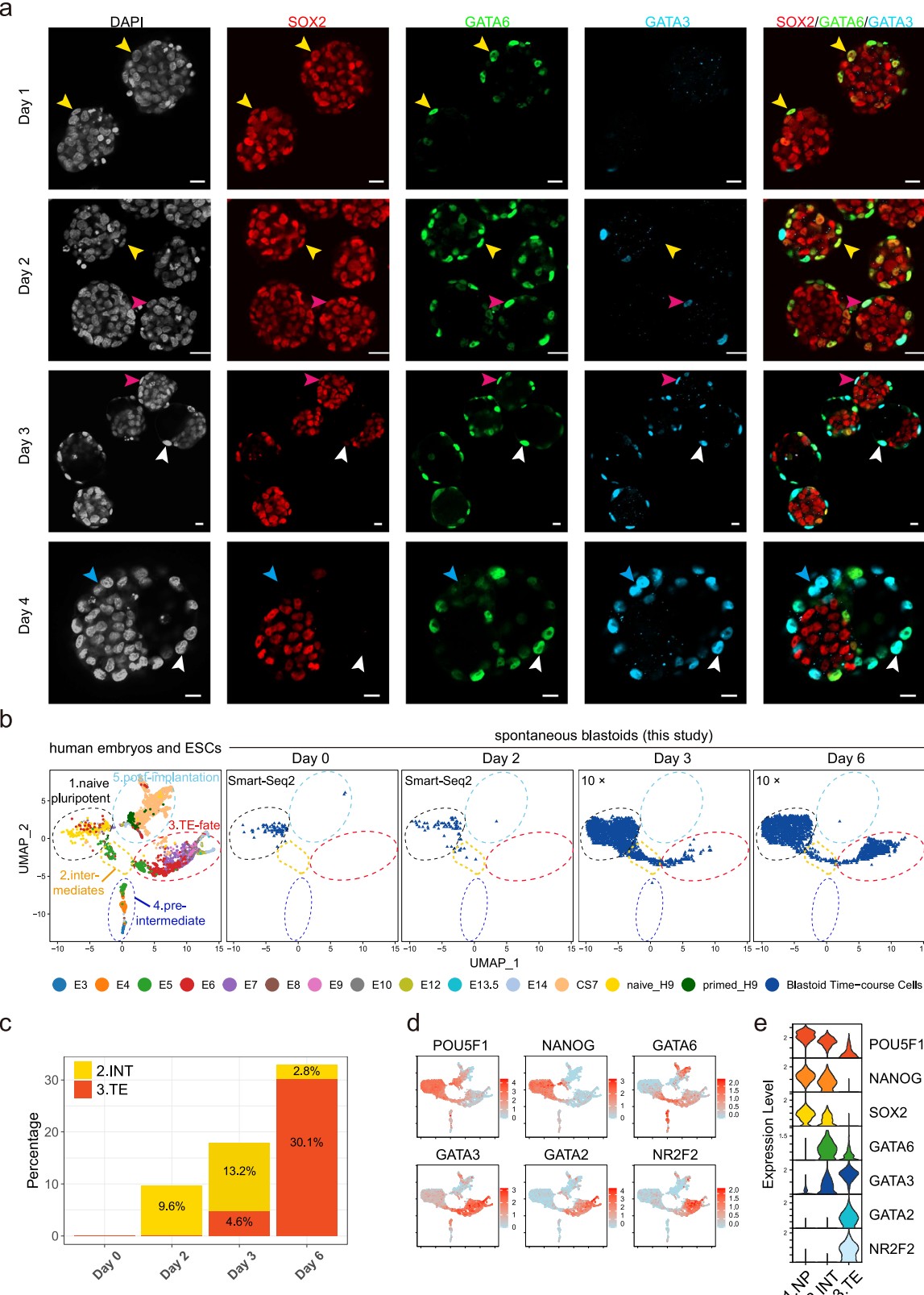

**Fig. 7 | Dynamics of cell fate specification during spontaneous blastoids formation. a** Representative time-course immunofluorescence staining images of day 1–4 aggregates/blastoids in AggreWell. Yellow arrows: SOX2⁺GATA6⁺GATA3⁻ cells; White arrows: SOX2⁻GATA6⁺GATA3⁺ cells; Pink arrows: SOX2⁺GATA6⁺GATA3⁺ cells; Blue arrows: SOX2⁻GATA6⁻GATA3⁺ cells; Scale bars, 20 μm. (*n* > 10). **b** Joint UMAP analysis and re-cluster of published human embryos/ESCs data (left) and day 0, 2, 3 and 6 spontaneous blastoid scRNA-seq data (right). **c** Contribution of cluster 2 and 3 to day 0, 2, 3 and 6 blastoid cells. **d, e** Expression levels of marker genes in cells and clusters. Source data are provided as a Source Data file.

feeders, we mainly focused on blastoids from AggreWell without feeders in this study.

For the individual colony blastoids generated from hniPSCs in 96-well U-bottom plate, the colonies with cavities were counted as blastoids. The blastoid formation efficiency was calculated as the number of blastoids per total number of picked colonies.

## Derivation of cell lines from human blastoids

Individual human blastoids were plated onto feeders by a mouth pipette and cultured in 5iLAF-I medium (for blastoid-nPSCs), RACL medium (for blastoid nEnd) or TSC medium (for blastoid TSCs) following previously described protocols[28,30,34]. Blastoids attached to feeders the other day and outgrowths could be observed. Colonies were formed and passaged on day6. Blastoid-nPSCs were cultured and passaged as we did with CI-hnESCs.

Blastoid nEnd cells were derived in RACL medium, and after 6 days of culture in RACL, they were passaged in NACL medium. The RACL medium was prepared containing: RPMI 1640 medium (Gibco, C22400500BT), 2 mM GlutaMAX, 2% B27 minus insulin (Gibco, A1895601), 100 ng/ml Activin A, 3 μM CHIR99021 and 10 ng/ml recombinant human LIF. The NACL medium was prepared containing: 1:1 of DMEM/F12 and Neurobasal, 1% N2 supplement, 2% B27 supplement, 2 mM GlutaMAX, 1% MEM Non-Essential Amino Acids, 0.5% penicillin-streptomycin, 0.1 mM 2-mercaptoethanol,100 ng/ml Activin A, 3 μM CHIR99021 and 10 ng/ml recombinant human LIF. Blastoid nEnd cells were passaged with Accutase every 6 days and 10 μM Y-27632 was supplied for 24 h after seeding.

Blastoid TSCs were derived and cultured in TSM containing: DMEM/F-12 supplemented with 0.3% BSA, 0.2% FBS (Gibco,10099141 C), 1% ITS-X supplement (Gibco, 51500056), 0.1 mM 2-mercaptoethanol, 2 mM GlutaMAX, 0.5% penicillin-streptomycin, 1.5 μg/ml l-ascorbic acid (Sigma, A8960), 1 μM SB431542 (Selleck, S1067), 50 ng/ml EGF (PeproTech, AF-100-15), 0.5 μM A83-01 (Selleck, S7692), 5 μM Y-27632, 2 μM CHIR99021 and 0.8 mM valproic acid (Selleck, S3944). The medium was changed daily, and the cells were passaged every 6 days using TrypLE Express (Gibco, 12604039).

## Differentiation of TSCs to EVT and ST like cells

Differentiation of blastoid TSCs to later stage trophoblast cells were performed following previously described protocols[34]. In brief, the blastoid TSCs were first dissociated with TrypLE Express, then seeded onto gelatinized plate and incubated at 37 °C for 60 min in normoxia to remove feeders. For EVT differentiation, the feeder-free TSCs were collected and resuspended in EVT basal medium supplemented with 4% KSR,100 ng/ml NRG1 (Cell signaling technology, 26941 S), and 2% Matrigel. $1 \times 10^5$ cells were seeded into a Geltrex-coated well for 6-well plate. On day 3, the medium was replaced with EVT basal medium supplemented with 4% KSR, reduced Matrigel (0.5%) and without NRG1. On day 6, the medium was replaced with EVT basal medium supplemented with 0.5% Matrigel to further remove KSR. On day 8, the cells were ready for downstream analysis.

For ST differentiation, feeder-free TSCs were collected and resuspended in ST medium. $1 \times 10^5$ cells were seeded into a Geltrex-coated well for 6-well plate. The medium was refreshed on day 3. On day 6, the cells were ready for downstream analysis.

The EVT basal medium comprised of DMEM/F12 supplemented with 0.1 mM 2-mercaptoethanol, 0.5% penicillin-streptomycin, 0.3% BSA, 1% ITS-X, 7.5 μM A83-01, 2.5 μM Y27632. The ST medium comprised of DMEM/F12 supplemented with 0.1 mM 2-mercaptoethanol, 0.5% penicillin-streptomycin, 0.3% BSA, 1% ITS-X, 2.5 μM Y-27632, 2 μM Forskolin (Sigma, F3917).

## Post-implantation development of spontaneous blastoids

Spontaneous blastoids are formed by refreshing 5iLAF medium daily for 5 days before they are subjected to post-implantation differentiation. Post-implantation blastoid culture was performed according to the protocol developed by Karvas et al.[49] with minor modifications.

Prior to post-implantation differentiation, 8-well chamber slide (Ibidi) or 96-well plate is pre-coated with 75% Cultrex (R&D, BME001) in DMEM/F12 for at least 1 h in the incubator (37 °C, 5% CO₂, 5% O₂). Post-implantation differentiation medium is also balanced for at least 1 h in the incubator (37 °C, 5% CO₂, 5% O₂).

To initiate post-implantation differentiation, Day5 blastoids that have an ICM and a typical TE cavity are manually picked with cut-tips or mouth-pipette tubes under a dissection microscope and carefully transferred to the pre-coated wells (3-5 blastoids/well for 8-well chamber slide and 1-2 blastoids/well for 96-well plate). On the next day, attachment of the blastoids is evaluated under a microscope. Once the TE appears to invade into the matrix and the blastoids lose mobility, the medium is carefully removed and replaced with freshly balanced medium. Minor morphological change of TE for over 48 h is considered failure of attachment and the samples will be suspended. Post-implantation differentiation medium is refreshed daily for the attached blastoids for 9 days.

Post-implantation differentiation medium is comprised of 1:1 Neurobasal:DMEM/F12, supplemented with 1x B27, 1x N2, 1x Glutamax, 1x NEAA, 1x sodium pyruvate, 0.1 mM β-mercaptoethanol, and 10 nM Estradiol. 3 mg/ml glucose is optionally added for differentiation beyond 7 days.

## Immunofluorescence staining

Blastoids were fixed with 4% paraformaldehyde (PFA, Beyotime, P0099) for 30 min at room temperature or overnight at 4 °C. Blastoids were then washed briefly with Dulbecco's phosphate-buffered saline (DPBS, BasalMedia, B210KJ) and permeabilized with 0.75% Triton X-100 (Sigma, T8787) in DPBS for 1 h using a mouth pipette, followed by blocking with blocking buffer (DPBS containing 3% BSA, 0.1% Triton X-100) at room temperature for 1 h. Then blastoids were incubated with primary antibodies diluted in blocking buffer at 4 °C overnight. After being washed for three times with DPBS containing 0.1% Triton X-100, blastoids were then incubated with Alexa Fluor (488, 555 or 647) conjugated secondary antibodies (Invitrogen) against the appropriated species at 1:1000 for 1 h at room temperature in the dark. After secondary antibody incubation, cell nuclei were stained with 4′,6-diamidino-2- phenylindole (DAPI, Sigma, D9542). Blastoids were transferred into small drops of DPBS onto glass-bottom dishes (NEST,801001) under mineral oil (Sigma, M8410) and imaged using Zeiss LSM800 or LSM880 laser scanning confocal microscopes. Using ZEN software, images were processed and diameter of each blastoid was measured. The details of antibodies and their dilutions are provided in Supplementary Data 3.

## Bulk RNA-seq preparation and sequencing

Total RNAs were collected and isolated from cells using RNAiso Plus (Takara, 9109). mRNA was purified from total RNA using poly-T oligo-attached magnetic beads. Sequencing libraries were generated using NEBNext®Ultra™ RNA Library Prep Kit for Illumina® (NEB, E7530L) following manufacturer's recommendations and index codes were added to attribute sequences to each sample. The library preparations were sequenced on an Illumina Novaseq platform at Novogene Co., Ltd and 150 bp paired-end reads were generated.

## Bulk RNA-seq data analysis

**Data pre-processing and quality control.** Raw reads of fastq format from this study were first processed through in-house perl scripts to obtain clean reads by removing reads containing adapter, reads containing ploy-N and low-quality reads from raw reads, while raw reads from public data GSE76970[36] and GSE133097[29] downloaded from NCBI Gene Expression Omnibus (GEO) were trimmed using Trim Galore

(v.0.6.4, http://www.bioinformatics.babraham.ac.uk/projects/trim_galore/) with default parameters to generate clean reads. All the downstream analyses were processed based on the clean data. Alignment, annotation, and gene expression quantification were subsequently performed with STAR[66] software (v.2.7.9a). References, including pre-built genome sequences and gene annotations, were downloaded from Ensembl GRCh38 release 104 (http://www.ensembl.org/Help/ArchiveList). Default parameters were used when implementing STAR workflow except the following: '--twopassMode Basic' for novel splice detection via two-pass method and '--quantMode GeneCounts' for gene expression quantification. The second columns of the output 'ReadsPerGene' tables were aggregated into a gene count table.

**Differential gene expression analysis and principal componence analysis.** Downstream analyses were performed in R 4.0.5 using DESeq2[67] and limma[68] packages. Data transformation, including normalization for library size and vst transformation, were performed according to the pipeline suggested by the development team of DESeq2 package (http://www.bioconductor.org/packages/release/bioc/vignettes/DESeq2/inst/doc/DESeq2.html#countmat). Differential expressed genes between CI-hnESCs and C-hnESCs were identified via DESeq function using the following cutoffs: adjusted $P$-value < 0.05 and the absolute value of Log2 Fold Change > 1. Batch effects between public data and the data from this study were corrected with function removeBatchEffect from limma package. Principle Componence Analysis (PCA) was subsequently performed on the batch-corrected data, using the top 500 most variable genes, via plotPCA function from DESeq2 package. The Volcano Plots were generated with EnhancedVocano[69] package, and the results of PCA were visualized with ggplot2[70] package (https://ggplot2.tidyverse.org).

## scRNA-seq library generation and sequencing
Routine cultured CI-hnESCs and day 2 aggregates were dissociated and sorted by flow cytometer for Smart-seq2 library construction and the libraries were sequenced on MGISEQ2000 (MGI Tech) platform to generate 100-bp paired-end reads. Day 3, day6 blastoids and the blastoids upon removing MEK/BRAF inhibition from Aggrewell were manually picked and polled with a mouth pipette, washed once with DPBS and then dissociated with 0.25% trypsin-EDTA (Gibco, 25300054) at 37 °C in 20%$O_2$ and 5% $CO_2$ incubator for 30 min. Attached cultured bE14 blastoids were first rinsed once with DPBS and then treated with cold DPBS and incubated in 4 °C for 10–20 min to allow breaking of the matrix. When the samples became mobile under a microscope upon gentle shaking, the bE14 blastoids were transferred to a clean tube using a cut-tip and dissociated with 0.25% trypsin-EDTA at 37 °C in 20%$O_2$ and 5% $CO_2$ incubator for 30 min. We then added 1 ml of DPBS containing 0.04% BSA and centrifuged the sample at 300 g for 5 min. Cell pellet was resuspended in DPBS containing 0.04% BSA and then filtered through a 40-μm cell strainer. Cell number was determined using Countstar Mira FL with AOPI fluorescence staining to assess cell viability. Single-cell libraries were generated following the manual instructions of 10X Genomics (Chromium Single Cell 3' Reagent Kits, V3.1). The libraries were sequenced either on an Illumina Novaseq platform at Novogene Co., Ltd to generate 150-bp paired-end reads (day 6 blastoids) or on MGISEQ2000 (MGI Tech) platform to generate 28-bp (R1) and 100-bp (R2) paired-end reads (day3 blastoids, the blastoids upon removing MEK/BRAF inhibition and the blastoids undergoing post-implantation development).

## scRNA-seq data analysis
**scRNA-seq data pre-processing and quality control.** Single-cell RNA-seq data of day 6 blastoids was pre-processed through Trimmomatic software[71] to remove adapters and to drop low quality reads or short reads below 26 bps. Other single cell RNA-seq data generated in this study and the reads from public data downloaded from E-MTAB-3929, GSE136447, E-MTAB-9388, GSE134571, GSE156596, GSE171820 and GSE177689[3,18,20,22,44–46] were downloaded and trimmed using Trim Galore coupled with FastQC (v.0.11.2) (https://www.bioinformatics.babraham.ac.uk/projects/fastqc/) or fastp software with default parameters. Alignment, annotation, PCR duplicate removal and gene expression quantification were subsequently performed using the STARsolo pipeline (https://github.com/alexdobin/STAR/blob/master/docs/STARsolo.md) with STAR[66] (v.2.7.9a). For Smart-seq2 data, trimmed reads were aligned to the reference and uniquely mapped reads were quantified with parameter '--soloType SmartSeq'. PCR duplicate removal was performed with parameter '--soloUMIdedup Exact'. 10× genomics data were processed via the same pipeline with parameter '--soloType Droplet, --soloCBstart 1, --soloCBlen 16, --soloUMIstart 17, --soloUMIlen 12, --soloBarcodeReadLength 0, --soloFeatures Gene Velocyto' with '3M-february-2018.txt' downloaded from cellranger website as whitelist file (https://github.com/10XGenomics/cellranger/blob/master/lib/python/cellranger/barcodes/translation/3M-february-2018.txt.gz). Count matrix of E10-E14 embryo data from Ai et al.[50] was kindly provided by the authors. Further analyses were performed in R 4.0.5 with Seurat[72] (v.4.1.0). For data generated in this study, quality controls were performed based on initial evaluation of per-cell quality control metrics. In particular, for the data-sets of spontaneous blastoids cultured in 5iLAF for 0 and 2 days (Smart-Seq2), cells with less than 2000 detected genes were filtered out; for the data-sets of day 3 and day 6 spontaneous blastoids, the blastoids upon removing MEK/BRAF inhibition as well as the blastoids undergoing post-implantation development (10X Genomics), cells with less than 2500, 4000, 2000 and 2000 detected genes respectively or with total read counts over 100000 or over 25%, 20%, 30% and 25% mitochondrial gene percentage were excluded. DoubletFinder[73] software was used to detect and filter hybrid cells. In later analysis of the data-set of day 6 spontaneous blastoids, a 'doublet' cluster was detected and excluded. For public data, cells with less than 2000 detected genes or >=20% mitochondrial gene percentage were filtered out. Genes detected in at least 3 cells were retained. Log-normalization, centering and scaling of the filtered count data were performed with NormalizeData and ScaleData functions sequentially and top 2000 highly variable genes were selected with FindVariableFeatures function.

**Dimensional reduction and clustering analysis.** Principal component analysis (PCA) was applied on the scaled data using RunPCA function in Seurat with default parameters based on the selected highly variable genes. A shared nearest neighbor (SNN) was constructed with the PCA coordinates using FindNeighbors function and were subsequently partitioned via Louvain algorithm implemented via FindClusters function at the resolution of 0.7 (day6 spontaneous blastoids) and 0.1 (blastoids upon removing MEK/BRAF inhibition). UMAP was implemented using RunUMAP function for visualization. The UMAP embeddings and cluster assignments computed in this section were later used in RNA velocity analysis.

**Differential gene expression analysis and computation of GSVA scores.** Wilcoxon Rank Sum test (two-sided) was performed on each cluster to identify differently expressed genes using FindAllMarkers function in Seurat with $p$-value adjustment via bonferroni correction. Cluster markers were filtered at a padj cutoff of 0.05. GO terms were enriched using DAVID website and were visualized with ggplot2 in R. Expression levels of selected markers shown in the violin-plots and feature-plots are log-normalized counts transformed with NormalizeData function and visualized respectively with VlnPlot and FeaturePlot function from Seurat package. Differentially expressed genes of relevant lineages (as were shown in Supplementary Fig. S6d) were identified using FindMarkers function in Seurat within batch at a padj cutoff of 0.05 and Log2FC cutoff of 1 (Supplementary Data 4).

Markers of Definitive endoderm and ExE endoderm were downloaded from Mackinlay et al.[52]. Then GSVA scores were computed using gsva function from GSVA package[74] with default parameters and normalized within blastoid identities the highest values across embryonic lineages to 1. Higher scores represent higher similarity of the selected blastoid identity with the corresponding embryonic lineages.

**Integrative analysis with public data.** We observed dataset-wise batch effect between our data and the public ones, and we accordingly implemented batch effect correction and data integration pipelines. In the following analysis, top 3000 most variable genes were involved and default parameters were used unless otherwise mentioned. For integrative analysis between Petropoulos, et al.'s embryos and our blastoids, we used Seurat CCA approach for anchors determination and datasets alignment. Features that were repeatedly variable across datasets were selected and anchors were identified using FindIntegrationAnchors function with following parameters: 'k.anchor = 3, k.filter = 180, k.score = 40'. The datasets were then integrated based on the identified anchors with IntegrateData function. Scaling and PCA were applied on the combined datasets and the integrated PCA coordinates were used as the input of the clustering and UMAP visualization workflow with the following parameter: 'dims = 1:30'. For the integration of all public data, fastMNN approach was adopted. Each dataset from different studies was considered as a batch. For visualization purpose, the time-course spontaneous blastoids data-sets were randomly sampled to no more than 2000 cell per batch (Fig. 7b, c, Source Data related with Fig. 7c). We only kept genes that were expressed in at least 5 cells in each of the datasets. Log-normalization was performed using computeSumFactors function from scran[75] package (v.1.20.1) and scaling normalization across batches was then performed with multiBatchNorm function in batchelor[76] package (v.1.8.1). The log-normalized, batch-effect corrected datasets were thereafter integrated using the fastMNN approach implemented via SeuratWrappers (v.0.3.0). The MNN low-dimensional coordinates were later used for graph-based clustering and visualization, which were implemented as described in previous sections using the following: 'varGene = 2000, reduction = "mnn", dims = 1:20' (Fig. 7b), 'reduction = "mnn", dims = 1:30, min.dist = 0.5' (Fig. 4d and Supplementary Fig. S4d–i) and 'reduction = "mnn", dims = 1:30, min.dist = 0.4' (Fig. 5c and Supplementary Fig. S6a–c) in the RunUMAP and FindNeighbors functions. Annotations of cells from blastoids upon post-implantation development (Fig. 5c) were initially defined based on clustering results within batch (at the resolution of 3) but then corrected combining the co-clustering results of the spontaneous blastiods with the embryonic data-sets, expression levels of conventional lineage markers and GSVA scores computed with DEGs of corresponding lineages (Fig. 5d, f and Supplementary Fig. 6).

**RNA velocity.** Spliced and unspliced matrices of reads were computed via starsolo pipeline using parameter '--soloFeatures Gene Velocyto'. We adopted UMAP embeddings and cluster assignments computed from previous section and used scVelo[77] package in Python 3.8.12 to compute RNA velocity with default parameters.

### Reporting summary
Further information on research design is available in the Nature Portfolio Reporting Summary linked to this article.

## Data availability
RNA-seq datasets generated in this study have been deposited in the NCBI GEO database under accession code: GSE252114. The raw data are publicly available. Previously published sequencing data that were reanalyzed here are available in the GEO or ArrayExpress under the accession codes E-MTAB-3929, GSE136447 and E-MTAB-9388 (scRNA-

seq data of E3-E7, D6-D14 and CS7 human embryos); Count matrix of E10-E14 embryo data from Ai et al.[50] was kindly provided by the authors; GSE134571, GSE156596, GSE171820 and GSE177689 (scRNA-seq data of blastoids reported in previous studies); and the sample GSM2041716, GSM2041717 (UCLA20 hnESCs) and GSM4721334, GSM4721335 (TJ-1# hnESCs). Source data are provided with this paper.

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

## Acknowledgements
We thank all colleagues in our laboratory for their assistance with the experiments and comments on the manuscript. We are grateful to Jianfeng Zhou in Tongji University for his assistance with cell culture. We appreciate Dr.Wenjuan Li and Dr.Miguel A. Esteban in Guangzhou Institutes of Biomedicine and Health for kindly providing STiPS O-XX1 cell line. This work was funded by the National Natural Science Foundation of China (32200586 to M.G.), China Postdoctoral Science Foundation (2022M712208 to M.G.) and by the Department of Science and Technology of Guangdong Province (2021ZT09Y233 to J.C.R.S. and M.M.). The J.C.R.S. laboratory is supported by Guangzhou Laboratory.

## Author contributions
J.C.R.S. supervised the study. M.G., C.C. and J.C.R.S. designed experiments, wrote manuscript, and analyzed data. M.G., C.C., J.W., X.W. generated the figures. J.W. and W.G. designed and performed computational analysis. M.G., C.C., and X.W performed experiments. A.G. and M.M. assisted with the live cell imaging. K.W. assisted with cell culture. Y.W. and S.G. provided cell lines and insightful comments. R.M.K. and T.W.T. assisted with the post-implantation differentiation of spontaneous blastoids and provided insightful comments.

## Competing interests
J.C.R.S., M.G., J.W., C.C., and K.W. submitted a patent application (202211174246.1) related to this work. Other authors declare no competing interests.
