## [Peer Review File · Nature Communications]

Self-renewing human naïve pluripotent stem cells
dedifferentiate in 3D culture and form blastoids
spontaneouslyReviewers' Comments:

Reviewer #1 (Remarks to the Author):

Guo and colleagues report the spontaneous formation of blastocyst-like structures from naive human pluripotent stem cells grown in suspension culture. The authors use indirect immunofluorescence to show that markers of epiblast and trophoctoderm are expressed in appropriate regions of the cystic structures that emerge efficiently following suspension culture of one naive cell line. scRNA-seq confirmed the identity of the two cell lineages through comparison with published studies of gene expression in pre-implantation embryos. The authors show that formation of blastoids in their system depends on the use of a specific GSK3beta inhibitor that resulted in higher expression of genes related to oxidative phosphorylation and extracellular matrix genes; blockade of oxidative phosphorylation led to inhibition of blastoid formation.

There is great interest in models of early human development derived from pluripotent stem cells. Several prior studies have shown that naive hPSC can give rise to blastocyst-like structures. These other studies have demonstrated formation of the three lineages of the pre-implantation blastocyst, epiblast, trophoblast and primitive endoderm. The other studies also documented properties of the trophoblast and primitive endoderm in great detail, and analysed further development in culture models of implantation and post-implantation development. By contrast, Guo et al. do not describe formation of primitive endoderm to any significant degree, and their characterization of the epiblast and trophoctoderm in their model is relatively superficial compared to the prior works. The findings of spontaneous formation of blastoids in suspension culture, and its enhancement by a different GSK3beta inhibitor, are interesting but need to be fully documented in more cell lines in order to understand how widely valid these observations really are.

Specific points:

L24 (and L64) naive pluripotent stem cells do not respond directly to lineage cues because they need to undergo a process of capacitation whereby they acquire competence to respond to these signals. This does not mean that their identity is unclear, a similar requirement applies to cells of the epiblast. See the work of Austin Smith on this topic.

L45 recent data from the Smith lab PMID: 36240776 argues against the notion that epiblast, trophoctoderm and primitive endoderm lineages emerge concurrently in the human. This recent work does not resolve the controversy fully but should be acknowledged.

L104 and following putative trophoblast cells should be characterized for further differentiation potential as should putative epiblast cells

L135 much further work would be required to demonstrate that the authors have actually derived trophoblast cells or extraembryonic cells from their blastoids. Characterization beyond immunostaining with a few markers is essential. Also the origin of these cells is unclear, it is possible they would have formed from naive cells under these conditions without any blastoid formation.

L 187 it is impossible to tell from Figure 3 a that the IM-12 improved the morphology of the cells.

L192 clarify what is meant here. How are the conditions described (5iLAF-C) different from those in

which spontaneous blastoid differentiation was observed (L77 and following)?

L202 Does UCLA20 spontaneously form blastoids?

L221 how do the authors know that the OXPHOS inhibition was not just toxic to the cells? Aggregates may have formed then died. Did the authors measure proliferation after addition of the inhibitor? The same consideration applies to removal of other naive culture components described in the next paragraph, maybe these conditions simply arrest cells altogether, and the observations say nothing specific about blastoid formation.

Figure 4 and Supplementary Figure 5-How do we know the cells staining for markers in 4c, or those not characterized in Supplementary Figure 5, are similar to human pre-implantation cells as shown in Figure 2? It is important to demonstrate how general these observations are and the characterization of the cells in Figure 4 and particularly Supplementary Figure 5 is rudimentary at best.

Reviewer #2 (Remarks to the Author):

In this study, Mingyue Guo et al. show that the TJ1 cell line cultured in a naive medium (5iLAF) can form structures resembling the blastocyst either in 2D or in 3D culture. Although the structures formed morphologically resemble the blastocyst, the time necessary for their formation is of 6 days instead of <4 for morula-to-blastocyst transition and previously established protocols (Yanagida et al., Kagawa et al.). The authors show that the effect of the medium is mainly driven by PD03 and Y27632, a fact that matches previous observations by Lo et al., Guo et al., Yanagida et al., and Kagawa et al. However, the scRNAseq data shows that the cells formed largely resemble the post-implantation stage, while overlapping as well, to some extent, with blastocyst-stage cells. Overall, the findings are largely overstated, especially the manuscript title and some paragraph titles. I don't think it represents an advance in the field. Although the findings related to the role OxFos could be interesting, the fact that they are tested in a model that does not reflect the blastocyst stage does not allow firm conclusions to be made.

The understanding of the state of the art in both blastoid and hhPSCs does not match my knowledge of the literature (see below).

The fact that the structures take 6 days to form when the time to develop from a human morula to a late blastocyst is 3.5 day shows that the proposed culture is not optimal. Of note, the protocols of Yanagida et al. and Kagawa et al. form blastoids within less than 4 days.

The single-cell sequencing analysis of the author's cells, when projected onto the most advanced reference map done by Sophie Petropoulos and Fredrik Lanner shows that the cells largely reflect the post-implantation stages (supplementary figure 3 d-e).

Considering the current state of the art, it is likely that the medium used by the authors leads to a slower lineage specification and morphogenesis as compared to human embryos and to established blasted protocol, which consequently leads to the appearance of largely differentiated cells. This is a major drawback of this protocol.

The title of this manuscript is incorrect (based on the scRNA seq data) and misleading as it makes a claim that is already well established while insinuating novelty.

Line 45: it is incorrect to say that there is still a debate on the sequence of lineage specification that leads to human blastocyst formation: This issue has been solved by linking pseudo-time analysis of scRNA-seq data with time-lapse imaging of annotated embryos [PMID: 34004179]. The sequence is similar between mouse and human: first the TE and ICM form, then the EPI and PrE form. These results have been confirmed by careful staining in the following biorxiv paper [doi.org/10.1101/2022.12.08.519626].

Line 50: The references are commentaries that are not necessary and should be replaced by the original research papers, as added at the end of the sentence.

Line 54: It should be added that naive hPSCs must be properly stimulated in order to recapitulate the sequence of lineage specification of blastocyst development and form blastocyst-stage-like cells. The reference to Yanagida et al. and Kagawa et al. should be added as the meta-study from Zhao et al. showed that the study of Yu and al. largely formed post-implantation-like cells.

Line 56: It should be noted that both pseudo-time analysis of the human blastocyst [PMID: 34004179] and dissociation/reconstruction experiments [PMID: 23257394] both showed that early human blastocysts contain cells that are still labile, undetermined and that maintain a capacity to form TE.

Line 62: This sentence should be more precise as hnPSCs largely resemble the blastocyst epiblast, and the 8-cell-like cells represent a very small subpopulation.

Line 64: This fact is not surprising in light of the assessed capacity of human ICM cells being able to form TE (see comments above).

Line 65: This fact can be simply explained by a stage-specific capacity to respond to the inductive signals necessary to form germ cells. There is no reason to find it cryptic as this is a classical development biology process of stage-specific competency.

Line 89: Naming them blastoids is unadapted, especially considering the results of the scRNAseq data.

Line 91: The efficiency is lower than the already established protocol by Kagawa et al. The tile should be changed as it implies an improvement on previous protocols that is not justified.

Line 96: The fact that structures resembling the blastocyst can form in 5iLAF medium is interesting, although not surprising as the medium contains PD03, a compound that has a major effect on TE specification as shown by Lo et al [PMID: 33831365] and Guo et al. [PMID: 33831366], as shown by the authors later in the manuscript (line 226). My interpretation is that the 3D configuration (aggregation) is likely to provide positional information (inner-outer cells) that, in conjunction with PD03 allows for the formation of TE-like cells. I would also suggest that the authors repeat this experiment with other cell

lines as this behaviour could be a particularity of the TJ1 line.

Line 104: Sox2 and Gata3 are not markers specific to the blastocyst stage. They are expressed during many stages of development of the epiblast and trophoblast lineages. Please check <https://petropoulos-lanner-labs.clintec.ki.se/app/shinyblastoids>

Line 121: These results are different from unpublished observations in multiple labs. I would recommend that the authors remain precautions while interpreting these experiments.

Line 148 and above: The scRNAseq data shows that the cells largely reflect a post-implantation stage. The title is not justified and the model is a step backward as compared to previously established protocols.

Reviewer #3 (Remarks to the Author):

In their manuscript titled Human naive pluripotent stem cells are functional blastocyst forming cells, Guo and colleagues observe spontaneous cavitation in their human naive pluripotent stem cell cultures. They sequence these structures and compare them to human early blastocyst and blastoid scRNA-seq datasets. Further, they show that oxidative phosphorylation is required for the formation of their blastoids. I do have some worries that they did not find primitive endoderm in their blastoids and therefore they may not model early human implantation as well as existing blastoid models already do. The results of this study are still interesting however and may have implications for the mechanisms driving trophectoderm differentiation and cavitation. See further comments below:

Please comment on why your scRNA-seq did not identify primitive endoderm like cells in your blastoids. If you look at the expression of more primitive endoderm markers genes than just GATA6 in your scRNA-seq, such as FOXA2 and SOX17, does that help? You should look at more marker genes in your immunofluorescence too, as GATA6 is expressed in your intermediates and early TE and by immunofluorescence it is often at the outer layer with the TE so that may just be early TE. In humans GATA6 is expressed earlier than in mouse in the embryo. Deriving nEND on RA1 from blastoids also is not really evidence of primitive endoderm as naive hPSCs will form nEND in that media too.

Correlation analysis with the Petropoulos dataset may help show how close your cells are to the embryo data, maybe with other blastoid datasets for comparison.

Does OXPHOS inhibition affect only blastoid generation in 5iLA or in other blastoid media too?

Around line 231 or elsewhere, please list and compare the components of 5iLA and previously used blastoid medias.

In figures 2 and S3 there seems to be some instances where text is overlapping or cut off

The sex of the cell lines used is not mentioned anywhere. Please include this.

Point-by-Point Response

Encouraged by the positive assessment and constructive comments made by the reviewers we have now performed an in-depth revision of our manuscript. In this we have performed significant additional experiments, conducted further computational analyses and re-wrote the manuscript. Together, we feel to have significantly improved our manuscript. We have also addressed each specific point raised by the reviewers, as detailed below. The reviewers' comments are in black and our replies are indicated in blue.

Reviewer #1 : Guo and colleagues report the spontaneous formation of blastocyst-like structures from naive human pluripotent stem cells grown in suspension culture. The authors use indirect immunofluorescence to show that markers of epiblast and trophoctoderm are expressed in appropriate regions of the cystic structures that emerge efficiently following suspension culture of one naive cell line. scRNA-seq confirmed the identity of the two cell lineages through comparison with published studies of gene expression in pre-implantation embryos. The authors show that formation of blastoids in their system depends on the use of a specific GSK3beta inhibitor that resulted in higher expression of genes related to oxidative phosphorylation and extracellular matrix genes; blockade of oxidative phosphorylation led to inhibition of blastoid formation.

There is great interest in models of early human development derived from pluripotent stem cells. Several prior studies have shown that naive hPSC can give rise to blastocyst-like structures. These other studies have demonstrated formation of the three lineages of the pre-implantation blastocyst, epiblast, trophoblast and primitive endoderm. The other studies also documented properties of the trophoblast and primitive endoderm in great detail, and analysed further development in culture models of implantation and post-implantation development. By contrast, Guo et al. do not describe formation of primitive endoderm to any significant degree, and their characterization of the epiblast and trophoctoderm in their model is relatively superficial compared to the prior works. The findings of spontaneous formation of blastoids in suspension culture, and its enhancement by a different GSK3beta inhibitor, are interesting but need to be fully documented in more cell lines in order to understand how widely valid these observations really are.

We thank the reviewer for the positive comments and constructive suggestions. In the revised manuscript we have now expand on how spontaneous blastoids are generated. By performing immunofluorescence and scRNAseq time course analysis we demonstrate that upon 3D culture hnPSCs at the periphery of the aggregates dedifferentiate, within 2 days, from an E6 embryo-like molecular identity into cells with a molecular identity similar to E5 intermediate embryo cells (see Figure 6 and replies below for further details). These intermediates, and not the starting hnPSCs, are the cells that subsequently undergo blastocyst cell fate specification/blastoid formation.

We now also demonstrate that hypoblast cell fate is specified in spontaneous blastoids upon the removal of BRAF and MEK signaling inhibition (see Figure 4 and supplementary Figure 5 and replies below for further details). We removed BRAF and MEK signaling inhibition at day 3, concurrently with cavitation formation, and analyzed spontaneous blastoids 1-2 days later (at day 4 and 5). Specified hypoblast cells match embryo hypoblast reference data, express expected markers and show correct location within spontaneous blastoids (Figure 4 and supplementary Figure 5).

We have now performed further computational analyses using additional reference datasets and new scRNAseq data for our spontaneous blastoids. This further confirms that our spontaneous blastoids map very close to pre-implantation embryo data and to also other blastoid data, further reinforcing that our spontaneous blastoids are molecularly similar to pre/peri-implantation human blastocyst (Figure 3, 5 and 6 and supplementary Figure 4. see also replies below).

We have now also performed differentiation of our spontaneous blastoids and these generate structures with complex morphology and that exhibit marker expression of post-implantation developmental stages further validating spontaneous blastoids as a model for human embryo development (see below for further details).

We have now also demonstrated that blastoid formation generated by an independent method, eHT method (Yu et al., BioRxiv, 2022), is also impaired by OXPPOS inhibition (Supplementary Fig.2b). We have now also demonstrated that the OXPPOS inhibitor we used does not affect cell proliferation during the time blastoid/cavitation is formed (Figure 2i). Together these results reinforce OXPPOS as an important parameter capacitating hnPSCs for blastoid formation.

Spontaneous blastoid formation was also validated into 2 additional hnPSC lines, including the widely used H9 cell line (please find this in Figure 5 and supplementary Figure 6 and also in the below replies).

Altogether, our new results further highlight spontaneous blastoids as being biologically significant and resembling the human blastocyst.

Specific points:

L24 (and L64) naive pluripotent stem cells do not respond directly to lineage cues because they need to undergo a process of capacitation whereby they acquire competence to respond to these signals. This does not mean that their identity is unclear, a similar requirement applies to cells of the epiblast. See the work of Austin Smith on this topic.

We agree with the reviewer and have now removed these sentences from the revised manuscript.

L45 recent data from the Smith lab PMID: 36240776 argues against the notion that epiblast, trophoderm and primitive endoderm lineages emerge concurrently in the human. This recent work does not resolve the controversy fully but should be acknowledged.

We thank the reviewer for pointing this study to us. We have now updated our introduction by including and citing recent works about early lineage specification, including the study suggested by the reviewer. We also support the reviewer indicated line of thought.

L104 and following putative trophoblast cells should be characterized for further differentiation potential as should putative epiblast cells

The putative trophoblast cells have now been characterized for further differentiation as recommended by the reviewer. By using the published protocol (PMID: 29249463), we evaluated the capacity of the derived putative trophoblast cells to differentiate to either extravillous trophoblast (EVT) or syncytiotrophoblast (ST) (Supplementary figure.2e-h and figure below). After cultured in EVT differentiation medium, villus-like

morphology emerged from the derived TSCs and quantitative PCR results showed these cells had upregulated EVT marker (*MMP2*) compared to the parental cells (Supplementary figure.2g). On the other hand, when cultured in ST medium, multinucleated cells could be observed, which is a feature of ST, and qPCR results showed upregulation of ST marker (*SDC1*) compared to the parental TSCs (Supplementary figure.2h). So far, we had evidenced for the expression of TSC markers in our derived TSCs and have now demonstrated the bipotential of our putative TSCs to differentiate into terminal fates (EVT and ST) morphologically and molecularly.

As for the differentiation potential of epiblast cells within the blastoid please see reply below.

L135 much further work would be required to demonstrate that the authors have actually derived trophoblast cells or extraembryonic cells from their blastoids. Characterization beyond immunostaining with a few markers is essential. Also the origin of these cells is unclear, it is possible they would have formed from naive cells under these conditions without any blastoid formation.

We have now further analyzed the derived trophoblasts cells from our blastoids. Apart from the previous immunostaining, we also evaluated the lineage specific gene expression of our derived TSCs. Quantitative PCR data showed TSC related genes *KRT7* and *TFAP2C* were upregulated in TSCs (Supplementary figure.2e-h and below). By using the published protocol (PMID: 29249463), we evaluated the capacity of the derived putative trophoblast cells to differentiate to either extravillous trophoblast (EVT) or syncytiotrophoblast (ST).

As for the origin of the cells competent to induce blastocyst cell fates/cavitation these have now been determined. The generation of spontaneous blastoids is not only rapid and efficient but also it does not involve media change enabling that way the study of the underlying biology of these with reduced variables. We now also report that 3D culture causes hnPSCs at the boundary of the aggregates to dedifferentiate, within the 2 days, from a E6 embryo-like naïve identity into E5 embryo-like intermediates (Figure 6b-e and below). It is these reprogrammed intermediates, and not hnPSCs directly, that are then competent to

undergo specification into blastocyst fates/blastoid formation. Please also note that there is no pre-differentiation in our hnPSC cultures (see scRNAseq data at day 0 in Figure 6b). Intermediates, and subsequent cell derivatives, arise exclusively upon 3D culture.

We also show that our spontaneous blastoids are similar to pre-implantation human blastocyst embryos and to other reported blastoids (Please see Figure 3e and f, Figure 6b and Supplementary Figure 4d-i. See also replies below for further details).

In addition, we now also show that spontaneous blastoids can specify hypoblast fate upon the removal of BRAF and MEK signaling inhibition (please see Figure 4 and replies below).

We now also show that the spontaneous blastoids progress in development in a rapid and sophisticated manner upon changes to the culture system (unpublished protocol from Thorold W. Theunissen lab). Spontaneous blastoids formed in AggreWell on day5, were harvested and transferred into a chamber slide precoated with a substrate. On day 9 spontaneous blastoids were collected and fixed for staining. Please note the complex and structured morphology and expression of TBXT, a marker of post-implantation development, in the exemplifying confocal section below. Purple indicates GATA3 expression and red SOX2.

L187 it is impossible to tell from Figure 3 a that the IM-12 improved the morphology of the cells.

We thank the reviewer for pointing this to us. We have now added two new panels to show the distinct and improved morphology of hnESCs cultured with IM-12 (please see Figure 2a and below).

C-hnESCs

CI-hnESCs

L192 clarify what is meant here. How are the conditions described (5iLAF-C) different from those in which spontaneous blastoid differentiation was observed (L77 and following)?

We observed spontaneous blastoid formation in standard 5iLAF medium which means IM-12 was used as GSK3 inhibitor (this is the GSK3b inhibitor used in the original 5iLAF medium formulation reported by Theunissen et al, 2014 PMID: 25090446 when generating hnPSCs). To distinguish the two, we called it 5iLAF-

C when using GSK3b inhibitor Chir99021 (used by some labs and in many other media formulations) and 5iLAF-I when using IM-12 (or just 5iLAF in other sections of the manuscript). No other difference exists between these media, that is the beauty of our system, the variables are quite simple. The details about 5iLAF-I and 5iLAF-C media are now also better described in the relevant results section.

L202 Does UCLA20 spontaneously form blastoids?

We followed the reviewer suggestion and have emailed Professor Amander Clark (clarka@ucla.edu) in different occasions in order to have access to the UCLA20 hnESC line. Unfortunately, we did not get a reply to our emails and do not know the reason for this.

However, we have tested spontaneous blastoid competence in two additional human pluripotent stem cell lines; the STIPS O-XX1 iPSC line and the widely used H9 ESCs. Importantly, both showed spontaneous blastoids formation competence in 3D culture using two different strategies, by colony picking into round bottom wells or/and in Agrewell plating (Figure 5 and Supplementary Fig.6).

L221 how do the authors know that the OXPPOS inhibition was not just toxic to the cells? Aggregates may have formed then died. Did the authors measure proliferation after addition of the inhibitor? The same consideration applies to removal of other naive culture components described in the next paragraph, maybe these conditions simply arrest cells altogether, and the observations say nothing specific about blastoid formation.

We thank the reviewer for raising this point. As the reviewer suggested, we have now measured cell proliferation upon OXPPOS inhibitor addition. Importantly OXPPOS inhibition did not visibly alter hnESC proliferation in 2D culture or aggregate size in 3D culture during blastocyst formation (Figure 2i, see also below, Supplementary Fig.3b). This result further supports OXPPOS as an underlying cause for hnPSCs capacitation to undergo spontaneous blastoid formation upon 3D culture.

As mentioned above we also tested if OXPPOS could be underlying blastoid formation in an independent method (Yu et al., BioRxiv, 2022). Similarly, we observed that OXPPOSi significantly impaired blastoid formation in this system (supplementary Fig.3c) These results further reinforce OXPPOS as an important variable for blastoid formation.

We have now also investigated the relevance of each 5iLAF medium component for cell proliferation. Apart from ROCK inhibition (ROCKi), absence of individual 5iLAF components impaired blastoid formation efficiency without affecting cell proliferation (Fig.1m, n and below). ROCKi Y-27632 is known to promote

cell survival suggesting that cell viability upon hnPSC dissociation and plating in suspension is not only a crucial parameter in our spontaneous blastoid assay but also in all other reported blastoid systems. These results further demonstrate that robust self-renewing culture conditions are integral for spontaneous blastoid generation.

Figure 4 and Supplementary Figure 5-How do we know the cells staining for markers in 4c, or those not characterized in Supplementary Figure 5, are similar to human pre-implantation cells as shown in Figure 2? It is important to demonstrate how general these observations are and the characterization of the cells in Figure 4 and particularly Supplementary Figure 5 is rudimentary at best.

We thank these comments. In previous Figure 4 (current Figure 5) we mostly showed blastocyst-like structures that are spontaneously generated from detached hniPSC colonies. These structures are in all similar to our spontaneous blastoids generated from suspended hnESC colonies (shown in Supplementary 1a-e). The data generated in Supplementary Figure 6b was from single cell dissociated hniPS single cells, which we have now also characterized these by immunofluorescence (Figure 5f and below). These spontaneous blastoids are also similar to hnESC spontaneous blastoids generated in the same way. These have a human blastocyst-like morphology, correct size, and both TE and EPI lineage marker expression.

Reviewer #2 : In this study, Mingyue Guo et al. show that the TJ1 cell line cultured in a naive medium (5iLAF) can form structures resembling the blastocyst either in 2D or in 3D culture.

We do appreciate the reviewer constructive comments and suggestions. These were very useful to help us improve our manuscript.

Although the structures formed morphologically resemble the blastocyst, the time necessary for their formation is of 6 days instead of <4 for morula-to-blastocyst transition and previously established protocols (Yanagida et al., Kagawa et al.).

This is a misunderstanding, our spontaneous blastoids (cavitation) are formed by day 3 (please see reply below for further details and also, supplementary videos, Figures 1c and Figure6a).

The authors show that the effect of the medium is mainly driven by PD03 and Y27632, a fact that matches previous observations by Lo et al., Guo et al., Yanagida et al., and Kagawa et al.

This is also a misunderstanding, what drives spontaneous blastoid formation is not PD03 or/and Y27632, it is increased OXPPOS combined with 3D culture (this is not to say that they are not required as we show that robust self-renewal conditions and cell survival are important for spontaneous blastoid generation). We now also show that hnESCs at the boundary of 3D aggregates dedifferentiate, within 2 days, to an earlier stage of embryonic development (E5 embryo-like) with intermediate characteristics, rendering these cells with the ability to then specify blastocyst cell fates and blastoid/cavitation formation (please see Figure 6 and replies below). In brief, there are 3 key aspects to the generation of spontaneous blastoids; 1- increased OXPPOS (provided by GSK3bi IM-12) which capacitates hnPSCs to form blastoids, 2- 3D culture which causes dedifferentiation of E6 embryo-like hnPSCs into E5 embryo-like embryo intermediates, 3- Intermediates, and not the starting hnPSCs, are competent to undergo blastocyst cell fate specification/blastoid formation. These 3 key findings are also very novel and of significance. We have now re-written the manuscript, and provide substantial new data, to make these points clearer. Please see also reply below for further details and new data.

However, the scRNAseq data shows that the cells formed largely resemble the post-implantation stage, while overlapping as well, to some extent, with blastocyst-stage cells.

While some TE fate cells show indeed progress towards post-implantation stages (in the previous version we had analyzed blastoids at only day 6) this is not the case for epiblast-like cells. We have mapped our new day 2-6 spontaneous blastoid scRNAseq data with embryo reference data from different stages (Figure 6b and below). Indeed, for the TE-like cells, some resemble E7/8 trophoblast, but we also have E5-E6 TE-like cells indicating that the TE part of our spontaneous blastoids continue to mature even in 5iLAF medium. As for our blastoid EPI-like cells those match embryo reference E6 epiblast cells which corresponds to pre-implantation stage.

As a whole our blastoid cells are located very close to the pre-implantation embryo cells and are molecularly similar to pre/peri-implantation embryo cells in terms of EPI and TE fate (please see also Figure 3 and Supplemental Figure 4 for further details). Our spontaneous blastoids are also very similar to other reported blastoids (Supplementary Figure 4). Please see also our replies below for further details.

We think that this misunderstanding may have been caused from having called embryo cells as Epiblast cells in former Figure 2e (now Figure 3e). Please note that the term “epiblast” in here corresponds to E6-7 naïve pluripotent epiblast embryo cells.

Overall, the findings are largely overstated, especially the manuscript title and some paragraph titles. I don't think it represents an advance in the field. Although the findings related to the role OxFos could be interesting, the fact that they are tested in a model that does not reflect the blastocyst stage does not allow firm conclusions to be made.

The understanding of the state of the art in both blastoid and hhPSCs does not match my knowledge of the literature (see below).

We have performed a substantial revision of our manuscript, following the reviewer suggestions, and have also re-written the manuscript to make our points clearer. Following the reviewer suggestion, we have also substantially revised our text on the state of the art of blastoids and hnPSCs. Please note that our new data demonstrating that hnPSCs dedifferentiate upon 3D culture into E5 embryo-like intermediates has implications about our understanding of hnPSCs properties and of how blastoids are formed (is not hnPSCs that differentiate into TE, it is instead the reprogrammed/dedifferentiated intermediate cells).

The fact that the structures take 6 days to form when the time to develop from a human morula to a late blastocyst is 3.5 day shows that the proposed culture is not optimal. Of note, the protocols of Yanagida et al. and Kagawa et al. form blastoids within less than 4 days.

This is a misunderstanding, our spontaneous blastoids are formed/cavitate at day 3 (not at day 6). Please see this in Figure 1c, in supplementary videos and in new Figure 6a (see also below). In new Figure 6a we performed an immunofluorescence time-course analysis which perfectly exemplifies the spontaneous blastoid kinetics. We have now stated this clearly in our manuscript. We think that the misunderstanding arose from the fact that we performed the staining's, scores and single cell sequencing analysis on blastoids at day 6 but this was not to mean that this was the timing of blastoid formation/cavitation.

The single-cell sequencing analysis of the author's cells, when projected onto the most advanced reference map done by Sophie Petropoulos and Fredrik Lanner shows that the cells largely reflect the post-implantation stages (supplementary figure 3 d-e).

Our scRNAseq data shows that spontaneous blastoid cells are almost exclusively close to pre-implantation embryo cells at the blastocyst stage (Figure 3e and f and panel below). Please note that the Epiblast label in Figure 3e and f represents pre/peri-implantation E5-E7 cells. We have now also mapped our day 6 blastoid scRNAseq, and new day 0, 2 and 3 data against embryo reference data broken by developmental time (Figure 6b). Indeed, for the TE-like fate, and in the day6 spontaneous blastoids, some cells map to E7/8 trophoblast, but we also have E5-E6 TE cells. This shows that the TE part of our spontaneous blastoids continue to develop in 5iLAF hnPSC self-renewing medium. EPI-like cells of day 6 spontaneous blastoids are still naïve and close to pre-implantation E6 embryo reference epiblast cells (Figure 3e and f and 6b). These results further reinforce that our spontaneous blastoids resemble the pre-implantation embryo.

Considering the current state of the art, it is likely that the medium used by the authors leads to a slower lineage specification and morphogenesis as compared to human embryos and to established blastocyst protocol, which consequently leads to the appearance of largely differentiated cells. This is a major drawback of this protocol.

We want to politely point out that our study is not a protocol. The role of the medium in the spontaneous blastoids regards the capacitation of hnPSCs to make blastoids spontaneously upon being in 3D culture. This capacitation is conferred by the GSK3bi IM-12 which increases the expression of genes involved in OXPPOS. Please note that by not requiring media change spontaneous blastoids represent a unique system to uncover properties of hnPSCs and on how blastoids are formed.

By making use of this unique system, we were now able to find that the 3D culture causes the dedifferentiation of E6 embryo-like hnPSCs, within 2 days, into E5 embryo-like intermediates. The intermediates are the cells competent to undergo specification into blastocyst fates/blastoid formation.

We also demonstrated that our spontaneous blastoids are similar to pre-implantation human embryo cells (Please see Figure 3e, 4d and 6b and other replies for further details). Spontaneous blastoid epiblast cells match/map E6 pre-implantation embryo reference epiblast cells

In addition, we now also show that spontaneous blastoids can specify hypoblast fate upon the removal of BRAF and MEK signaling inhibition. We also show that the spontaneous blastoids progress in development, generating complex morphologies and expression of TBXT, a marker of post-implantation development (please see example in the response to reviewer 1).

The title of this manuscript is incorrect (based on the scRNA seq data) and misleading as it makes a claim that is already well established while insinuating novelty.

The concept of self-renewing hnPSCs making blastoids spontaneously is novel. We have now also edited the title of the manuscript to better represent the revised manuscript findings. Our manuscript is now entitled: "Self-renewing Human naïve pluripotent stem cells dedifferentiate in 3D culture and form blastoids spontaneously". We now show that hnPSCs at the boundary of 3D hnPSC aggregates dedifferentiate to an intermediate/earlier stage of embryonic development. It is those cells that appear upon hnPSCs going into 3D culture that have the ability to specify blastocyst fates and cavitation/blastoid formation.

Line 45: it is incorrect to say that there is still a debate on the sequence of lineage specification that leads to human blastocyst formation: This issue has been solved by linking pseudo-time analysis of scRNA-seq data with time-lapse imaging of annotated embryos [PMID: 34004179]. The sequence is similar between mouse and human: first the TE and ICM form, then the EPI and PrE form. These results have been confirmed by careful staining in the following biorxiv paper [doi.org/10.1101/2022.12.08.519626].

We agree with the reviewer and have removed those suggested hypotheses from our manuscript. We have now re-written this as per suggested by the reviewer.

We have now also determined that hnPSCs generate blastoids spontaneously via these reprogramming to an earlier stage of embryonic development, when in 3D culture, and not due to hnPSCs per se.

We have now updated our introduction by including the suggested references about the early lineage specification.

Line 50: The references are commentaries that are not necessary and should be replaced by the original research papers, as added at the end of the sentence.

We appreciate the reviewer pointing this out to us. We have now corrected this.

Line 54: It should be added that naïve hPSCs must be properly stimulated in order to recapitulate the sequence of lineage specification of blastocyst development and form blastocyst-stage-like cells.

We thank the reviewer comments. In our case, hnPSCs form blastoids spontaneously in 3D culture with 5iLAF self-renewing medium, indicating hnPSCs are able to recapitulate lineage specification spontaneously. We have now demonstrated that this is due to hnESCs reprogramming to an earlier developmental stage when in 3D culture. The reprogrammed cells resemble E5 embryo-like intermediates that have the capacity to specify blastocyst fates and cavity/blastoid formation. We think this is very novel and of great significance to the field.

The reference to Yanagida et al. and Kagawa et al. should be added as the meta-study from Zhao et al. showed that the study of Yu and al. largely formed post-implantation-like cells.

We appreciate this comment. The suggested references (Yanagida et al. and Kagawa et al.), representing pre-implantation like cells, were both added (See supplementary figure 4d-I and below). They located very

close to pre-implantation embryo and to each other as expected and also very close to our blastoids, further reinforcing that our blastoids are molecularly similar to pre/peri-implantation embryo as other blastoids do in terms of EPI and TE fate.

Line 56: It should be noted that both pseudo-time analysis of the human blastocyst [PMID: 34004179] and dissociation/reconstruction experiments [PMID: 23257394] both showed that early human blastocysts contain cells that are still labile, undetermined and that maintain a capacity to form TE.

We agree with the reviewer and have corrected this.

Line 62: This sentence should be more precise as hnPSCs largely resemble the blastocyst epiblast, and the 8-cell-like cells represent a very small subpopulation.

We thank the reviewer for this comment and have now deleted this sentence. We have now also determined that it is the surrounding cells of 3D hnPSC aggregates that dedifferentiate from an E6 embryo-like naïve pluripotent identity into an E5 embryo-like intermediate molecular cell identity and that these intermediates are the responsible for blastocyst cell fate specification/cavitation formation (and not by any initial cells within self-renewing hnPSCs as it can be seen in the scRNAseq data in Figure 6b for day 0 cells).

Line 64: This fact is not surprising in light of the assessed capacity of human ICM cells being able to form TE (see comments above).

We have removed the sentence the reviewer indicates as our new data demonstrates that it is not hnPSCs per se that differentiate into TE/make the blastoids but instead the reprogramed hnPSCs that arise in the aggregates early on upon 3D culture. We find these findings novel and surprising and of significance to stem cell and developmental biology.

Human ICM cells being able to form TE is not surprising indeed as they will likely contain intermediates (cells in which fate has not yet been specified). However, the spontaneous generation of blastoids, and its underlying biology, from an already specified cell fate and in self-renewing conditions is very surprising.

Line 65: This fact can be simply explained by a stage-specific capacity to respond to the inductive signals necessary to form germ cells. There is no reason to find it cryptic as this is a classical development biology process of stage-specific competency.

In light of our new findings we now know that it is the hnPSCs reprogramming into E5 embryo-like cell intermediates representing an earlier stage of embryonic development that makes these competent to then undergo blastocyst cell fates/blastoid formation and not cues directing hnPSCs directly into a TE fate. This is not only surprising but very different from any current hypothesis.

Line 89: Naming them blastoids is unadapted, especially considering the results of the scRNAseq data.

We have to strongly disagree with the reviewer statement. Our blastoids not only show morphological features but also specification of TE fate matching embryo and independent blastoid reference data (Figure 3e and f and 6b and Supplementary figure 4d-i). We now also show that if relieved from BRAF and MEK signaling inhibition spontaneous blastoids readily specify hypoblast fate. In addition, our blastoids can subsequently further develop and generate sophisticated structures and molecular marker expression associated with human post-implantation development. Please also note that the focus of our study is on the observation of spontaneous blastoid generation and the further investigation into what causes it and how they are formed. Please also note that our study is not about developing a new blastoid protocol. We re-wrote the manuscript and clarified the scope of our study to make this clearer.

Line 91: The efficiency is lower than the already established protocol by Kagawa et al.

We show that spontaneous blastoid generation is rapid (it occurs within 3 days) (please see supplementary videos and Figure 6a) and with an efficiency above 50% (Figure 1f). This is not dissimilar from what Kagawa et al reported and is higher than reported in other studies (Sozen et al., Nature Communications, 2021; Liu et al., Nature, 2021; Yu et al., Nature, 2021). In addition, we report that dedifferentiation and subsequent TE specification occurs in all of 3D hnESC aggregates (Figure 6a) independently if these cavitate or not. Importantly we are reporting that these are spontaneous blastoid which occurred solely as a result of 3D culture, efficiency is not the point of our study.

The title should be changed as it implies an improvement on previous protocols that is not justified.

We have now changed the title of our manuscript to better represent the findings of our revised manuscript. Please also note that the focus of our study is on the report of spontaneous blastoid generation and investigation into what causes it and on how they are formed. Please also note that our study is not about developing a new blastoid protocol.

Line 96: The fact that structures resembling the blastocyst can form in 5iLAF medium is interesting, although not surprising as the medium contains PD03, a compound that has a major effect on TE specification as shown by Lo et al [PMID: 33831365] and Guo et al. [PMID: 33831366], as shown by the authors later in the

manuscript (line 226). My interpretation is that the 3D configuration (aggregation) is likely to provide positional information (inner-outer cells) that, in conjunction with PD03 allows for the formation of TE-like cells. I would also suggest that the authors repeat this experiment with other cell lines as this behavior could be a particularity of the TJ1 line.

We appreciate the reviewer comments and now have the data to answer this. PD03 (MEK/ERK signaling inhibition) is integral to the self-renewal of hnPSCs and robust self-renewal culture conditions are required for spontaneous blastoid formation. Please also note that we do not change medium composition reducing that way the variables involved in blastoid formation. What we found as underlying spontaneous blastoid formation was: 1- Capacitation of hnPSCs to form blastoids provided by increased OXPHOS, 2- 3D culture which caused the dedifferentiation of E6 embryo-like hnPSCs into E5 embryo-like intermediates, 3- Intermediates cells, not hnPSCs, having competence to undergo blastocyst cell fate specification/blastoid formation.

Please also note that 3D configuration is causal for the generation/induction of intermediate cells (not simply to sort cells) which are then the cells competent to undergo blastocyst cell fate specification/blastoid formation. This can clearly be seen in the scRNAseq time course analysis in Figure 6b. Intermediates are specified upon 3D culture and not before.

Spontaneous blastoid formation was also further validated into 2 independent hnPSC lines, including the widely used H9 cell line (Figure 5 and supplementary Figure 6).

Line 104: Sox2 and Gata3 are not markers specific to the blastocyst stage. They are expressed during many stages of development of the epiblast and trophoblast lineages. Please check <https://petropoulos-lanner-labs.clintec.ki.se/app/shinyblastoids>

We thank the reviewer for pointing this issue with our sentence. The reviewer is absolutely correct in stating that SOX2 and GATA3 are expressed during other stages of development and what we meant to say was within the blastocyst context (not in terms of the entirety of development). We have now rephrased this sentence. Please note that in addition to stainings we performed single cell sequencing analysis which we compared to embryo and to pluripotent stem cell reference data (Figure 3, 4 and 6; supplementary Figure 4). This confirmed unequivocally the molecular identity of our cells.

Line 121: These results are different from unpublished observations in multiple labs. I would recommend that the authors remain precautions while interpreting these experiments.

We thank the reviewer for these comments and we have taken them into account when revising our manuscript. We have not determined the cause for our cells not being able to generate blastoids following the PXGL/PALLY method. However, by communicating with other labs that derive and maintain hnPSCs using similar culture conditions to ours this is also an issue. However, when culturing our cells in PXGL they make spontaneous blastoids upon switching to 5iLAF medium.

In the revised manuscript we could generate blastoids following the recent Jun Wu's lab method (Yu et al., BioRxiv, 2022) (this lab uses a similar culture condition to ours to maintain hnPSCs self-renewing) (Supplementary Figure 3c and below). Importantly the cells of our spontaneous blastoids overlap with the

cells of blastoids from other labs, including those using PXGL/PALLY method. Please note that we are not challenging the results from any lab and will rephrase the mentioned sentence to avoid misunderstanding.

As a result of the time course single cell sequencing analysis we now determined that the cells at the periphery of the hnPSC aggregates dedifferentiate from an E6 embryo-like naïve pluripotent molecular identity into an E5 embryo-like intermediate molecular cell identity. It is these cells that dedifferentiated, and not the cells with a E6-like naïve pluripotent identity, that are responsible for cavitation formation and blastocyst cell fate specification. We were surprised by these results and they do change our initial concept/interpretation. These findings do not challenge the interpretation about the molecular identity of hnPSCs. Instead, it uncovered a new and surprising property of hnPSCs which is their ability to dedifferentiate upon being in 3D culture.

We thank the reviewer's comment. In the revised manuscript, we have performed a time-course immunofluorescence analysis during spontaneous blastoid formation. We confirmed that spontaneous blastoids self-assembled in 4 days gradually: spherical aggregates formed at day 1, TE-like cells were specified at day 2 and cavitation started at day 3. These results are different even from already published observations and indicate that hnPSCs are able to form blastoids in 3D and self-renewing culture with no need for any differentiation inducement. This highlights the significance of 3D culture in human blastoids production and provides a new platform to explore human early embryo development.

Line 148 and above: The scRNAseq data shows that the cells largely reflect a post-implantation stage. The title is not justified and the model is a step backward as compared to previously established protocols.

We have to politely disagree with the reviewer. Our scRNAseq data shows that blastoid cells are largely close to pre-implantation embryo cells (Figure 3e and f, 4d and e, Figure 6b and also above responses).

Please note that our study reports on the discovery that hnPSCs can make blastoid spontaneously and then on the investigation into what causes this and on how they are formed (we have an efficient system and media change is not a variable). Our findings suggested OXPHOS as a cause capacitating hnPSCs to generate spontaneous blastoids upon 3D culture. Then we found that hnPSCs at the periphery of 3D aggregates dedifferentiate into cells with a molecular identity similar to E5 intermediate embryo cells. These are then competent to undergo blastocyst cell fates/blastoid formation. Our findings are novel and have significant implications to our knowledge about hnPSCs and about blastoid formation. We are very thankful for the reviewer's comments and used these to improve our manuscript.

Reviewer #3 : In their manuscript titled Human naïve pluripotent stem cells are functional blastocyst forming cells, Guo and colleagues observe spontaneous cavitation in their human naïve pluripotent stem cell cultures. They sequence these structures and compare them to human early blastocyst and blastoid scRNA-seq datasets. Further, they show that oxidative phosphorylation is required for the formation of their blastoids. I do have some worries that they did not find primitive endoderm in their blastoids and therefore they may not model early human implantation as well as existing blastoid models already do. The results of this study are still interesting however and may have implications for the mechanisms driving trophoderm differentiation and cavitation. See further comments below:

We thank the reviewer for the positive and constructive comments. In the revised manuscript we now expand on how spontaneous blastoids are generated. By performing scRNAseq and immunofluorescence time course analysis we demonstrate that upon 3D culture hnPSCs at the periphery of the aggregates dedifferentiate, within 2 days, into cells with a molecular identity similar to E5 intermediate embryo cells. These intermediates subsequently undergo blastocyst cell fate specification/blastoid formation (please see Figure 6 and other replies).

We now also demonstrate that hypoblast cell fate is specified in spontaneous blastoids upon the removal of BRAF and MEK signaling inhibition. We removed it at day 3, concomitantly with cavitation, and analyzed it 1-2 days later. Specified hypoblast cells match embryo hypoblast reference data, express expected markers and show correct location within spontaneous blastoids (please see images below, Figure 4 and supplementary Figure 5).

We have now also performed differentiation of our spontaneous blastoids and these generate structures with complex morphology and exhibit marker expression of post-implantation developmental stages (please see example in the response to reviewer 1).

Please comment on why your scRNA-seq did not identify primitive endoderm like cells in your blastoids. If you look at the expression of more primitive endoderm markers genes than just GATA6 in your scRNA-seq, such as FOXA2 and SOX17, does that help?

The reviewer raises a very important point which we have now addressed in the revised manuscript. Our study was previously focused on the reporting of spontaneous blastoid formation and on the causes underlying this. Also, in the embryo cavitation precedes hypoblast cell fate specification. We were also aware, from our previous work "Nichols and Silva et al 2009, Development", that MEK/ERK signaling inhibition blocked PrE fate. Consistently with this, upon the removal of BRAF and MEK signaling inhibition, which we did at day 3 and concomitantly with cavitation, we observed GATA4+/SOX17+ hypoblast specification within 1-2 days in our spontaneous blastoids. Importantly, these appear at the expect location (see below, Figure 4 and Supplementary Figure 5). All the markers the reviewer suggested were also detected by either immunofluorescence or/and by scRNAseq analysis (see below, Figure 4 and Supplementary Figure 5b and d).

Supplementary Figure 5. Spontaneous blastoids specify hypoblast-like cell fate.

You should look at more marker genes in your immunofluorescence too, as GATA6 is expressed in your intermediates and early TE and by immunofluorescence it is often at the outer layer with the TE so that may just be early TE.

We thank the reviewer for these comments. In agreement with our new scRNA-seq (Figure 4e), we do find GATA4 and SOX17 protein expression in our spontaneous blastoids upon the removal of BRAF and MEK signaling inhibition (Figure 4c and Supplemental Figure 5b and d and above reply images). Please see also above replies for further details.

In humans GATA6 is expressed earlier than in mouse in the embryo. Deriving nEND on RA/L from blastoids also is not really evidence of primitive endoderm as naive hPSCs will form nEND in that media too.

We agree with the reviewer comment and now provide the above described datasets.

Correlation analysis with the Petropoulos dataset may help show how close your cells are to the embryo data, maybe with other blastoid datasets for comparison.

We thank the reviewer for this comment. We have plotted our data with Petropoulos et al dataset using the reannotation from Meistermann et al (PMID: 34004179) and we found that our blastoid cells are largely close to pre-implantation embryo (Figure 3e and f, Figure 4d, Figure 6b, supplementary figure 4 and images below). We also have compared our data with other blastoids datasets and these results further confirm naive EPI-like and TE-like cell identity in our spontaneous blastoid (Supplementary Fig.4e-h). In addition, we performed correlation analysis as was suggested (see heatmap below displaying spearman correlation coefficient, which were computed using batch effect corrected expression levels of top 3000 HVGs) against embryo data (Petropoulos et al. and Xiang et al. datasets) as well as other blastoid datasets (Kagawa et al. and Yanagida et al.). The results suggest that our blastoids show a considerable degree of similarity with embryo data as other blastoid datasets do, with our EPI-like and TE-like cells showing high similarity with human pre/peri-implantation EPI and TE cells.

d Human embryos (Petropoulos et al., 2016; Yanagida et al., 2021; Xiang et al., 2020; Tyser et al., 2021)

e Blastoids (This study)

f Blastoids (Liu et al., 2021)

g Embryonic-like sac (Zheng et al., 2019)

h Blastoids (Yanagida et al., 2021)

i Blastoids (Kagawa et al., 2022)

Does OXPPOS inhibition affect only blastoid generation in 5iLA or in other blastoid media too?

We thank the reviewer for this important comment. With our cell lines we were unable to repeat the established PXGL/PALLY blastoid protocols (but PXGL cultured cells can generate spontaneous blastoids in 5iLAF self-renewing medium). (Supplementary Fig.2a). However, we could generate blastoids following the recently reported eHT protocol (Yu et al., BioRxiv, 2022). We think this may be due to the fact that Yu et al derive and maintain hnPSCs in similar culture conditions to ours. Importantly, when OXPPOS inhibitor was applied, blastoid generation efficiency in the eHT protocol was also significantly impaired (Supplementary Fig.2c and below image). This suggests that OXPPOS may be critical for successful blastoid generation in general. We now have also demonstrated that the OXPPOS inhibitor we used does not affect cell proliferation during the time blastoid/cavitation is formed (Figure 2i and image above). Together these results reinforce OXPPOS as an important parameter capacitating hnPSCs for blastoid formation.

Around line 231 or elsewhere, please list and compare the components of 5iLA and previously used blastoid medias.

We used 5iLAF medium including MEK inhibitor **PD0325901**, GSK3 inhibitor **IM-12**, BRAF inhibitor SB590885, SRC inhibitor WH-4-023, ROCK inhibitor **Y-27632**, recombinant human LIF, Activin and FGF2 to generate blastoids spontaneously.

Yu et al used 5iLA(including **Y-27632**) for 12 hours, then hypoblast differentiation medium (FGF2, Activin A, ChIR99021) for 3 days and then trophoblast differentiation medium (ITS-X, **PD0325901**, A83-01, SB590885, WH-4-023, **IM-12**, CHIR99021, SB431542, human LIF, EGF, l-ascorbic acid and VPA) for 6 days to induce lineage specification and blastoids self-organization.

Yanagida et al used **PD0325901**, A83-01 and **Y-27632** for 44 hours and then A83-01 alone for 18 hours to induce blastoids from PXGL hnPSCs.

Kagawa et al used **PD0325901**, A 83-01, 1-oleoyl lysophosphatidic acid sodium salt (LPA), hLIF and **Y-27632** for 2 days and then LPA and Y-27632 for another 2 days to induce human blastoids.

We have listed the components of 5iLAF clearly in Materials and Methods. Components of other established protocol are easily found in each of the mentioned studies.

PD0325901 (MEK/ERK signaling inhibition) is integral to the self-renewal of hnPSCs and robust self-renewal culture conditions are required for spontaneous blastoid formation. Please also note that we do not change medium reducing that way the variables involved in our system compared to that of others. What we found to underlie spontaneous blastoid formation was the: 1- Capacitation of hnPSCs to form blastoids provided by increased OXPPOS, 2- 3D culture which caused the dedifferentiation of E6 embryo-like hnPSCs, at the

boundary of aggregates, into E5 embryo-like intermediates, 3- Intermediates cells, not hnPSCs directly, having competence to undergo blastocyst cell fate specification/blastoid formation.

In figures 2 and S3 there seems to be some instances where text is overlapping or cut off

We appreciate this comment and have now corrected this.

The sex of the cell lines used is not mentioned anywhere. Please include this.

The main cell line used in this study, TJ-1#, is male. The other two human cell lines, STiPS O-XX1 iPSCs and H9 ESCs, are both female. We do now state this info in the revised results and Materials and Methods sections.

REVIEWER COMMENTS

Reviewer #1 (Remarks to the Author):

The revised study of Guo et al. represents a considerable improvement on the original manuscript and addresses most of the criticisms of Reviewers 1 and 2. The authors have identified an E5 cell like intermediate as the origin of the blastoid structure, they have performed scRNA-seq to validate identities of the cell populations, and they provide evidence that the trophoblast-like cells have the capacity to undergo further development into extravillous trophoblast and syncytiotrophoblast. The authors report validation of the protocol with additional pluripotent stem cell lines, and demonstrate that modification of the protocol can produce some cells with hypoblast-like character.

These additional findings strengthen the study. However, the data on hypoblast formation are still not convincing. The immunofluorescence micrographs only show a few cells bearing the expected markers (these are not always in the expected positions), Figure 3 does not define the hypoblast population well, and Figure 4 d contains less than 10 cells that would fit in this category. The authors should provide more convincing data, perhaps by refining the protocol, to demonstrate this third key blastocyst lineage is formed robustly, or show further expansion of this lineage at later stages. It is true that hypoblast is a minor population in some other blastoid models but this protocol seems to generate very few compared to others. The authors should at least acknowledge this as a limitation. The findings on the role of OXPHOS are novel and interesting, but the authors should relate this to known features of human embryo metabolism derived from IVF studies. The micrograph of a later stage blastoid in the rebuttal document (TBXT staining) is somewhat confusing from a morphological point of view; exactly what stage of development this image represents is unclear and the evidence that the blastoid can progress beyond pre-implantation stages is not overly convincing.

In summary this study has been greatly improved and contains some novel observations. Although the work still has limitations, in this evolving field it is important to report new findings that might lead to further refinement of protocols for embryo modeling.

Reviewer #3 (Remarks to the Author):

I commend the authors on the vast improvements they have made to their submission. They have done well addressing my criticisms. A few minor comments on the new draft:

Figure 1k: 3 blastoids seems like a low number to count. There is a clear trend, but counting more blastoids would increase the confidence in your results. Also, I assume there was a criteria to be included in these counts. Was there more of a criteria than having an ICM and cavity as was used for quantifying efficiency? Please state this if so.

Figure 1n: What is the meaning of having this graph split into two parts? Is it just to improve readability or is there a reason that is not explained in the figure legend?

Figure 2c: having your color labels in a different order than your x-axis makes it harder to read

Figure 2f: please state in the figure legend what red vs yellow and green vs blue means.

Figure 3d: Why are there so many early EPI compared to other lineages? This seems unexpected compared to embryos, other blastoids, and figure 1k. I think this is an important point to address.

Figure 5C: As other blastoid protocols have needed to be optimized between different cell lines, this may be the case here as well. I think just mentioning this may be enough, as further optimization would defeat the "spontaneous" nature of your results

Line 306: "Importantly our day 0 hESCs did not show pre-existing cell differentiation", in figure 6b, there are d0 cells in the post-implantation cluster, would they not count as differentiated? Please clarify or change this claim

Additionally, I think that the paper could use another pass for grammar, as there were a few areas that were awkwardly worded.

REVIEWER COMMENTS

Reviewer #1 (Remarks to the Author):

The revised study of Guo et al. represents a considerable improvement on the original manuscript and addresses most of the criticisms of Reviewers 1 and 2. The authors have identified an E5 cell like intermediate as the origin of the blastoid structure, they have performed scRNA-seq to validate identities of the cell populations, and they provide evidence that the trophoblast-like cells have the capacity to undergo further development into extravillous trophoblast and syncytiotrophoblast. The authors report validation of the protocol with additional pluripotent stem cell lines, and demonstrate that modification of the protocol can produce some cells with hypoblast-like character.

We sincerely thank the reviewer for the supportive comments and for all the guidance throughout the review/revision process which helped us improve our manuscript.

These additional findings strengthen the study. However, the data on hypoblast formation are still not convincing. The immunofluorescence micrographs only show a few cells bearing the expected markers (these are not always in the expected positions), Figure 3 does not define the hypoblast population well, and Figure 4 d contains less than 10 cells that would fit in this category. The authors should provide more convincing data, perhaps by refining the protocol, to demonstrate this third key blastocyst lineage is formed robustly, or show further expansion of this lineage at later stages. It is true that hypoblast is a minor population in some other blastoid models but this protocol seems to generate very few compared to others. The authors should at least acknowledge this as a limitation.

We thank the reviewer for the suggestions which we have now addressed. Briefly:

- 1- We found that if the BRAF/MEK signalling inhibitors are removed at day 2, instead of day 3, this significantly increases the number of hypoblast cells per blastoid (please find this data and full response below). However, as indicated below, this also impacts blastoid formation efficiency.
- 2- We have also performed the 3 day 5iLAF spontaneous blastoid assay, followed by 2 day 3iLAF (without BRAF/MEK signaling inhibitors), in an independent human naïve pluripotent cell line (STiPS O-XX1 hniPSCs). This further confirmed that spontaneous blastoids generate hypoblast cell fate upon the removal of BRAF/MEK signaling inhibition (please find below this new dataset).
- 3- We have now performed the single cell RNA seq analysis of day 5 5iLAF spontaneous blastoids differentiated for 9 days using the recently published Karvas et al, 2023 (PMID: 37683602) method. As described below spontaneous blastoids progress into post-implantation fates and show a great alignment with human embryo reference single cell RNA seq datasets. This includes the generation of hypoblast cell derivatives, namely, visceral endoderm/Yolk sac endoderm and anterior visceral

endoderm. Together, these results show that spontaneous blastoids are also a useful post-implantation embryo-like model and further demonstrate that upon removal of BRAF/MEK signalling inhibition, in this case only from day 5 spontaneous blastoids, hypoblast fate is specified and cell derivatives generated. Please find this new data and its full description in the response to the reviewer last comment.

- 4- As the reviewer points out, studies on the blastoid system as a whole show under representation of the hypoblast lineage and that was also the case with our spontaneous blastoid model. Although not the focus of our study, we now acknowledge this as a potential limitation for the application of spontaneous blastoids as post-implantation embryo-like models (Line 435) and point out the need to improve hypoblast cell fate specification in order to refine this.

(continuation of point 1) To attempt improving hypoblast cell fate specification in our spontaneous blastoid model we have now removed MEK and BRAF inhibitors from day 2. This resulted in a significant increase of hypoblast cell number, to around 8 hypoblast cells per blastoid (see dataset below and supplementary Figure 5e-i). However, because cavitation in spontaneous blastoids depends on complete self-renewing medium, and this occurs mainly at day 3, removing BRAF/MEK signaling inhibitors at day 2 impacted negatively blastoid generation efficiency. This data has now been included in the re-revised manuscript.

(continuation of point 2) We have now also analyzed hypoblast cell fate specification in an independent human naïve pluripotent cell line (hniPSC line). The hniPSC line was converted from primed pluripotent iPSCs, using exclusively 5iLAF-I medium, and this shows similar spontaneous blastoid formation efficiency compared to hnESCs. Importantly, when under the spontaneous blastoid condition for 3 days (in 5iLAF) and then in 3iLAF for 2 days, hypoblast cell fate specification was also observed (see below and Figure 6g to k). These results further confirm that the spontaneously generated blastoids specify hypoblast fate upon removal of BRAF/MEK signaling inhibition.

The above newly generated results have now been added to the revised manuscript.

The findings on the role of OXPHOS are novel and interesting, but the authors should relate this to known features of human embryo metabolism derived from IVF studies.

We thank the referee for this excellent suggestion. Indeed, gene ontology analysis of single-cell RNA transcriptomes of human in vitro fertilized pre-implantation embryos showed that at the morula and blastocyst stages mitochondria/Oxidative phosphorylated genes are among the most significantly upregulated group of genes (PMID: 23934149 and PMID: 23892778). Oxygen consumption in the early human embryo at the morula stage was also found to correlate with faster progression to blastocyst stage (PMID: 32740687 and PMID: 28190213) and blastocysts consume high levels of oxygen mediated by high expression of OXPHOS genes (PMID: 29414683). We have now added this to the discussion section of our manuscript (Line409).

The micrograph of a later stage blastoid in the rebuttal document (TBXT staining) is somewhat confusing from a morphological point of view; exactly what stage of development this image represents is unclear and the evidence that the blastoid can progress beyond pre-implantation stages is not overly convincing.

(continuation of point 3) We appreciate the reviewer's point and have now asked if our spontaneous blastoids model can also develop and molecularly differentiate into post-implantation cell lineages (see below and Figure 5 and Supplementary Figure 6). We performed stainings and single cell RNA seq analysis of day 5 5iLAF spontaneous blastoids differentiated for 9 days using the recently published Karvas et al, 2023 (PMID: 37683602) method. As described below we found that spontaneous blastoids develop into post-implantation lineages and that they also closely align with counterpart post-implantation human embryo reference single cell RNA seq datasets. Please find below the new data and the description of this:

To address if spontaneous blastoids have the potential to progress into post-implantation development we differentiated these following the recently published method by Karvas et al (PMID 37683602) with some minor modifications. In brief, we simulated natural implantation by culturing our spontaneous blastoids on Cultrex-coated 8-well slides (Fig.5a). Attachment of the blastoids was rapid as evidenced by the outgrowth of trophoblast-like cells within 24h (Fig.5b).

Figure 5. Spontaneous blastoids have post-implantation developmental potential.

These morphological changes indicated blastoids functionally nidate and the EPI lineages differentiate. To

delineate the progression of embryonic and extraembryonic lineages of our blastoids, we collected and profiled the differentiated spontaneous blastoids 9 days after attachment at bE14 (Fig.5c-l and Fig.S6). With cross-reference to the pre- and post-implantation embryo data (E3-E19) [Petropoulos et al., 2016, PMID: 27662094; Xiang et al., 2020, PMID: 31830756; Ai et al., 2023, PMID: 37460804; Tyser et al., PMID: 34789876], the bE14 differentiated spontaneous blastoids showed clear alignment to reference post-implantation human embryo lineages and were best matched to the E14-19 dataset (Fig.5c).

C

d

Transcriptomic analysis revealed that blastoid mesoderm (bMES), definitive endoderm (bDE) and hemogenic endothelium (bHE) already emerged at the time point which overlapped to E16-19 dataset, indicating accelerated progression from primitive streak (bPS) (Fig.5c and Fig.S6a). The faster developmental pace was also observed in other embryo-like model systems [Weatherbee et al., 2023 PMID: 37369347; Ai et al., 2023, PMID: 37460804]. Immunostaining confirmed the presence of confined streak-like structures comprising T-expressing cells (Fig.5e).

Hypoblast derivatives (visceral endoderm/yolk sac endoderm (bVE/YSE) and anterior visceral endoderm (bAVE)) were also identified in our transcriptomic analysis, even though our spontaneous blastoids show impaired HYP commitment until the removal of BRAF/MEK signalling inhibition (Fig.3). Because bVE/YSE, bAVE and definitive endoderm (bDE) shared a plethora of endodermal markers (GATA4, GATA6, SOX17, LEFTY1) (Fig.5d and Fig.S6b), we questioned the annotation of extraembryonic endoderm lineages. To this end, we selected the DEGs between DE and extraembryonic endoderm identities from the embryo data and scored the annotated lineages from our differentiated blastoids against the embryo counterparts.

Gene set variation analysis (GSVA) demonstrated that our bDE like cells bore a higher similarity to the embryo DE identity while bVE/YSE and bAVE were more similar to the embryo extraembryonic identities (Fig.5f). In addition, we found evidence of secreted CER1 proteins in the extracellular regions of GATA4+ cell clusters (Fig.5g), suggesting that these cells may potentially be recapitulating the role of AVE cells as a signaling center to counteract BMP signaling [PMID: 32724077].

In view of extraembryonic tissue development, we found both trophoblast descendants, cytotrophoblasts (bCTBs) and syncytiotrophoblasts (bSTBs), and amnion-like cells (bAM) present in our UMAP analysis (Fig.5c and Fig.S7c). bAM cells showed highest resemblance to embryo amnion cells (Fig.5d and Fig.S7d). Of note, the reduced number of trophoblast-like cells in sequencing analysis compared to imaging was probably due to a technical issue because our blastoids were steadily attached and these lineages were deeply intruded into the matrix, as a result these cells could be lost while we harvested the samples for sequencing. Yet, bCTBs and bSTBs in our samples showed elevated expression of classic trophoblast markers GATA2, GATA3, NR2F2, TFAP2C (Fig.5d), and fluorescent staining results showed outgrowth of GATA3+ trophoblast like cells with apparent larger nucleus (Fig.5h and i). Presence of putative bSTBs was supported not only by sequencing results but also by CGB staining (Fig.5h). Potential EVT precursors were also found as evidenced by the presence of EVT marker HLA-G in GATA3 expressing cells (Fig.5i). Of note, we did not identify EVTs in the sequencing analysis. However, their frequency is rare at the equivalent embryonic developmental time point [Gauster et al., 2022, PMID: 35661923].

Supplementary figure 6. Spontaneous blastoids have post-implantation developmental potential

Together, these results suggest that spontaneous blastoids are also a useful post-implantation embryo-like model and confirm that these are competent to segregate extraembryonic endodermal lineages.

In summary this study has been greatly improved and contains some novel observations. Although the

work still has limitations, in this evolving field it is important to report new findings that might lead to further refinement of protocols for embryo modeling.

Once again, we are very grateful for the reviewer supportive comments and for all the helpful suggestions that led us to significantly improve our manuscript.

Reviewer #3 (Remarks to the Author):

I commend the authors on the vast improvements they have made to their submission. They have done well addressing my criticisms.

We thank the reviewer for all the effort in helping us improve our manuscript and for the positive assessment of our revision.

A few minor comments on the new draft:

Figure 1k: 3 blastoids seems like a low number to count. There is a clear trend, but counting more blastoids would increase the confidence in your results. Also, I assume there was a criteria to be included in these counts. Was there more of a criteria than having an ICM and cavity as was used for quantifying efficiency? Please state this if so.

Thank you for these comments. We have now quantified TE(Gata3+) vs ICM cells in eight additional spontaneous blastoids (total n=11) (updated Figure 1k and dataset below). The additional scores provide further robustness to this dataset.

.

Spontaneous blastoid lineage cell quantification.

(A) Immunofluorescence of 5 of the scored blastoids (1- 5 samples).

(B) Scores of all quantified blastoids.

(C) Violin plot of (B). n = 11 blastoids. Updated Figure.1k in the revised manuscript.

We defined a spontaneous blastoid as a blastocyst-like structure that contains an inner cell mass and a well-defined cavity. We now clearly state this in both the manuscript and in the Methods section (line 548). This criterion to evaluate blastoid formation efficiency is also the same as that in published blastoid studies.

Figure 1n: What is the meaning of having this graph split into two parts? Is it just to improve readability or is there a reason that is not explained in the figure legend?

Figure 1n is based on two independent experiments because we didn't have enough cells for all groups when we performed the first experiment. In each experiment we set a parallel control, cells cultured in 5iLAF. We have now made this clear in the figure legend. Having this dataset as top and bottom panels also improves the readability.

Figure 2c: having your color labels in a different order than your x-axis makes it harder to read

We thank the reviewer for pointing this out. We have now corrected this.

Figure 2f: please state in the figure legend what red vs yellow and green vs blue means.

Response: Upregulated genes are shown in yellow and red, and downregulated genes are shown in green and blue. We just show some representative genes in red (mostly OXPHOS associated genes) and in blue (mostly differentiation associated genes) for highlighting. We have now stated this clearly in the figure legend.

Figure 3d: Why are there so many early EPI compared to other lineages? This seems unexpected compared to embryos, other blastoids, and figure 1k. I think this is an important point to address.

Thank you for raising this comment. By immunofluorescence (Figure 1j, 1k) we could definitely conclude that there are more TE-like cells than EPI-like cells in our spontaneous blastoids and that the numbers for each lineage were also comparable to embryos and to other blastoid systems. However, we did find that more EPI-like cells are present in the 10x single cell RNA sequencing data. That is to say that there is a technical issue, likely occurring during spontaneous blastoid single cell dissociation and during the procedure of the capture of the single cells. TE-like cells are likely more fragile/sensitive to these procedures. A recent published study (Oldak B. et al., PMID: 37673118, Figure.6a) also highlighted this problem for similar stage embryo models. Therefore, we think that the total number of each lineage should be counted by immunofluorescence rather than by single cell RNA sequencing data. We do now mention this issue in the revised manuscript (line 230).

Figure 5C: As other blastoid protocols have needed to be optimized between different cell lines, this may be the case here as well. I think just mentioning this may be enough, as further optimization would defeat the "spontaneous" nature of your results.

We thank the reviewer for this comment and we agree with it. However, we could expand our findings to the two additional cell lines we tested, namely STiPS O-XX1 naïve iPSCs and H9 naïve ESCs. In the case of the hniPSCs, we have now also done side by side experiments and the spontaneous blastoid formation efficiency was found to be similar to hnESC (please find above, in the response to reviewer one, the description of this new dataset and also in Fig.6g-k).

Line 306: "Importantly our day 0 hESCs did not show pre-existing cell differentiation", in figure 6b, there are d0 cells in the post-implantation cluster, would they not count as differentiated? Please clarify or change this claim

The reviewer is correct and we have now amended this sentence. By this sentence our intention was to say that there are no pre-existing TE-like cells in day 0 starting hnESCs.

Additionally, I think that the paper could use another pass for grammar, as there were a few areas that were awkwardly worded.

Thank you for this suggestion.

REVIEWERS' COMMENTS

Reviewer #1 (Remarks to the Author):

The authors have made substantial and effective efforts to address the remaining concerns with the first revision of their manuscript. These include the addition of experimental data to address formation of extraembryonic endoderm, and additional data on the potential for post-implantation development of the model. In the revision the authors acknowledge limitations where they exist and put their data on metabolism in the context of prior work on pre-implantation embryos. The manuscript has been improved through two rounds of revision and represents an important contribution to the expanding body of work on human embryo models.